# Automatic and Structure-Aware Sparsification of Hybrid Neural ODEs with Application to Glucose Prediction

**Bob Junyi Zou**
Institute for Computational and Mathematical Engineering
Stanford University
Stanford, CA 94305
`junyizou@stanford.edu`

**Lu Tian**
Department of Biomedical Data Science
Stanford University
Stanford, CA 94305
`lutian@stanford.edu`

## Abstract

Hybrid neural ordinary differential equations (neural ODEs) integrate mechanistic models with neural ODEs, offering strong inductive bias and flexibility, and are particularly advantageous in data-scarce healthcare settings. However, excessive latent states and interactions from mechanistic models can lead to training inefficiency and over-fitting, limiting practical effectiveness of hybrid neural ODEs. In response, we propose a new hybrid pipeline for automatic state selection and structure optimization in mechanistic neural ODEs, combining domain-informed graph modifications with data-driven regularization to sparsify the model for improving predictive performance and stability while retaining mechanistic plausibility. Experiments on synthetic and real-world data show improved predictive performance and robustness with desired sparsity, establishing an effective solution for hybrid model reduction in healthcare applications.

## 1 Introduction

Hybrid modeling methods are receiving increased attention from the healthcare community because they combine inductive bias from mechanistic models with the flexibility of neural networks. These methods often prove especially valuable in small-data regimes—commonly found in healthcare, and medicine—by outperforming both fully black-box and purely white-box approaches in terms of predictive performance, robustness and interpretability (Ahmad et al., 2018; Du et al., 2019; Mohan et al., 2019; Rackauckas et al., 2020; Yazdani et al., 2020; Hussain et al., 2021; Karniadakis et al., 2021; Qian et al., 2021; Sottile et al., 2021; Zou et al., 2024).

In the field of dynamical system modeling, a significant category of hybrid approaches builds on neural ordinary differential equations (neural ODEs) (Haber & Ruthotto, 2017; Chen et al., 2018; Kidger, 2021), which arose from the insight that deep residual neural networks can be formulated as continuous-time dynamical systems (Rico-Martinez et al., 1992; Weinan, 2017). Neural ODEs are well-suited for modeling dynamical systems as they offer continuous-time representations with latent dynamics that integrate seamlessly into modern machine learning pipelines with automatic differentiation, making them both scalable and flexible. More recently, researchers have adapted neural ODEs to incorporate domain-knowledge-informed relational inductive bias, often derived from mechanistic models or causal graphs. This hybrid style—sometimes termed Graph Neural ODE (Poli et al., 2019), Neural Causal Model (Xia et al., 2021), Neural State Space Modeling (Hussain et al., 2021) or Mechanistic Neural ODE (MNODE) (Zou et al., 2024)—ensures mechanistic plausibility and interpretability while taking advantage of the flexibility of neural networks, thereby improving model performance and robustness in data-scarce settings.

While hybrid neural ODEs show competitive performance in various healthcare and medical applications such as cardiovascular simulation (Grigorian et al., 2024; Salvador et al., 2024), epidemic forecasting (Sottile et al., 2021; Huang et al., 2024), disease progression and survival analysis (Dang et al., 2023; Xiang et al., 2024), treatment effect estimation (Gwak et al., 2020; Zou et al., 2024) and pharmacology (Qian et al., 2021; Hussain et al., 2021), one persistent challenge in deploying them in

practice is model reduction. Mechanistic models in physiology and medicine tend to become excessively large in attempts to capture wide-ranging and complex dynamics (e.g., delays, heterogeneities, multi-compartment processes, etc.) and may contain dozens of latent states despite only a handful of observable states. For instance, the state-of-the-art model for human carbohydrate-insulin-glucose dynamics has more than 20 latent states, even though it only uses 2 input variables and less than 5 observable state variables (Visentin et al., 2018). After hybridization, the added flexibility of neural components may render some latent states unnecessary or even detrimental when training data are scarce, as redundant states can significantly increase model variance, leading to over-fitting and undermining the benefits mechanistic models promise.

Traditional model reduction approaches in biochemistry—such as timescale separation (Michaelis & Menten, 1913; Johnson & Goody, 2011) and quasi-steady-state approximations (Schauer & Heinrich, 1983; Bothe & Pierre, 2010)—often require deep domain expertise or extensive trial-and-error. On the other end of the spectrum, data-driven graph-based model reduction offers a pathway to solve this problem. In many healthcare application domains, mechanistic ODEs can be represented as reaction networks or directed graphs, where nodes denote system states and edges denote interactions (Hodgkin & Huxley, 1952; Holz & Fahr, 2001; Smith et al., 2004; Canini & Perelson, 2014; Man et al., 2014). In recent years, many solutions have emerged from the graph neural network (GNN) community for general graph pruning, using approaches such as topology-based node/edge selection (Spielman & Srivastava, 2008; Liu et al., 2023), learning-based sub-graph sampling (Wang et al., 2019; Zeng et al., 2019; Zheng et al., 2020), or optimization-based graph sparsification (Li et al., 2020; Jiang et al., 2021; 2023). However, these reduction methods are typically data-driven and agnostic of any domain knowledge, and thus do not necessarily preserve key mechanistic structures or constraints. Furthermore, non-gradient-based reduction methods (e.g., greedy search) can be prohibitively costly in computation for large, high-dimensional ODE systems. As a result, a gap remains for computationally efficient solutions that reduce model complexity while preserving the mechanistic integrity and improving predictive performance for hybrid neural ODEs.

In this paper, we address this challenge by introducing a hybrid, gradient-based algorithm for automatic state/edge selection and structure optimization in MNODEs. Our approach combines domain-knowledge-informed graph modification with a mix of $L_1$ and $L_2$ regularization that encourages graph sparsity to efficiently reduce model complexity. The graph modification step draws insights from classical reduction methods and graph theory to constrain the search space to mechanistically plausible sparse graphs that retain key topological structures. Meanwhile, the regularization step allows data-driven, gradient-based graph pruning during training, making the reduction process computationally efficient and adapted to observed data. By combining both mechanistic and data-driven elements, our reduction scheme integrates the best of both worlds and is particularly well-suited for modeling complex dynamical systems in healthcare and medicine with limited data. Through extensive experiments on both synthetic and real-world data, we demonstrate that our algorithm outperforms other reduction strategies for MNODEs and also surpasses unreduced MNODEs and widely used black-box sequence models in terms of predictive performance and robustness using less parameters. These findings highlight a promising path toward more efficient and effective hybrid modeling solutions—particularly in settings where high-quality data are scarce and model stability is crucial.

## 2 METHODOLOGY

### 2.1 Preliminary

**Mechanistic controlled ODE system** We define a mechanistic controlled ODE system $M$ as a 4-tuple $M = (S, X, F, S_0)$:

1. $S = \{s_1, \ldots, s_n\}$ is the set of state variables with cardinality $n$, and $s_i(t) : [0, +\infty) \to \mathbb{R}$ is a real-valued function of $t$ representing the value of state $s_i$ at time $t$.

2. $X = \{x_1, \ldots, x_m\}$ is the set of exogenous input variables with cardinality $m$, and $x_j(t) : [0, +\infty) \to \mathbb{R}$ is a function of $t$ representing the value of input $x_j$ at time $t$.

3. $F = \{f_1, \ldots, f_n\}$ is the set of real-valued functions of $S, X$ and $t$ that describe the system's temporal evolution:

$$\frac{ds_i(t)}{dt} = f_i(S_{\text{pa}(i)}(t), X_{\text{pa}(i)}(t), t),$$

where $S_{\text{pa}(i)} \subseteq S$, $X_{\text{pa}(i)} \subseteq X$ are subsets of state and input variables, respectively, on which the derivative of $s_i$ with respect to $t$ depends, i.e., they are "parents" of $s_i$.

4. $S_0 = \{s_1(0), \ldots, s_n(0)\}$ is the set of initial conditions.

In addition, we can further split the state variable set into two disjoint subsets:

$$S = \text{observable states } S_{\text{obs}} \sqcup \text{latent states } S_{\text{lat}}.$$

Observable states are variables in the system that can be directly measured through experiments or sensors. These are the quantities that can be collected and tracked over time. On the other hand, latent states are variables that are not directly accessible but still believed to play a role in system dynamics.

**Directed graph representation of mechanistic ODE** We define the directed graph representation of $M = (S, X, F, S_0)$ as a directed graph $G_M = (V_M, E_M)$, whose node set and edge set are defined in the following way:

$$V_M = S \cup X = \{s_1, \ldots, s_n, x_1, \ldots, x_m\},$$

$$(s_j, s_i) \in E_M \iff s_j \in S_{\text{pa}(i)}; \text{ and } (x_k, s_i) \in E_M \iff x_k \in X_{\text{pa}(i)}.$$

Specifically, $(s_j, s_i) \in E_M$ means that the value of $s_j(t)$ influences the "direction" of $s_i(t)$. Similarly, $(x_k, s_i) \in E_M$ means that the value of $x_k(t)$ influences the "direction" of $s_i(t)$. Note that we allow self loops–we can have $(s_i, s_i) \in E_M$, if $ds_i(t)/dt$ depends on $s_i(t)$ in the system. In the rest of the paper, we will use the following definitions:

**Relaxed directed acyclic graph (RDAG)**: We define a relaxed directed acyclic graph to be a directed graph with no directed cycles, except for self-loops. Note that the directed graph representations of mechanistic ODE systems are in general **NOT** RDAG.

## 2.2 Prediction task: time series forecasting with dynamic covariates

The main task we are interested in is to predict the future trajectory of observable state variables, given their observed history and both past and future exogenous input signals. More precisely, given:

(1) Past context: $S_{\text{obs}}^{\text{P}} = \{S_{\text{obs}}(t_k)\}_{k=-p}^{0} \in \mathbb{R}^{(p+1) \times |S_{\text{obs}}|}, X^{\text{P}} = \{X(t_k)\}_{k=-p}^{-1} \in \mathbb{R}^{p \times m}$, where $t_{-p} < \cdots < t_{-1} < t_0 = 0$ are a set of discrete time stamps at which observations of $S_{\text{obs}}$ and $X$ are collected, and $t_0$ is the beginning of the prediction window;

(2) Future inputs: $X^{\text{F}} = \{X(t_k)\}_{k=0}^{q-1} \in \mathbb{R}^{q \times m}$, where $0 = t_0 < t_1 < \cdots < t_q$ are future prediction time stamps in the prediction window;

we want to predict $S_{\text{obs}}^{\text{F}} = \{S_{\text{obs}}(t_k)\}_{k=1}^{q} \in \mathbb{R}^{q \times |S_{\text{obs}}|}$, the future value of the observable states.

**Data:** The observed data consist of copies of $\{S_{\text{obs}}^{\text{P}}, X^{\text{P}}, S_{\text{obs}}^{\text{F}}, X^{\text{F}}\}$ from multiple instances and the objective is to use observed data to train an algorithm prospectively predicting observable state values in new instances based on history, $\{S_{\text{obs}}^{\text{P}}, X^{\text{P}}, X^{\text{F}}\}$.

## 2.3 Model architecture: mechanistic neural ODE (MNODE)

At a high level, MNODE follows the encoder-decoder sequence modeling paradigm, in which the encoder takes in historical context and output an initial condition estimate of the latent states in the system and the decoder rolls out predictions based on the initial condition and future inputs:

$$\text{Encoder}(S_{\text{obs}}^{\text{P}}, X^{\text{P}}) = \hat{S}_{\text{lat}}(0), \quad \text{Decoder}(\hat{S}(0), X^{\text{F}}) = \hat{S}_{\text{obs}}^{\text{F}},$$

where $\hat{S}(0) = (S_{\text{obs}}(0), \hat{S}_{\text{lat}}(0)) \in \mathbb{R}^n$ is an initial condition estimate. In general, MNODE is compatible with any choice of encoder layer as long as the encoder can produce a reasonable estimate of the initial condition of the system. For the decoder layer, given the directed graph representation of the mechanistic ODE system $G_M$, future exogenous inputs $X^{\text{F}}$ and an initial condition estimate $\hat{S}(0) \in \mathbb{R}^n$, MNODE initializes node features in $G_M$ as $S^{t_0} = \hat{S}(0), X^{t_0} = X(0)$, and evolve state node features over time using a set of feed-forward neural networks $\{\text{NN}_i\}_{i=1}^{n}$ structured by $G_M$ :

$$\frac{ds_i(t)}{dt} = \text{NN}_i(S_{\text{pa}(i)}(t), X_{\text{pa}(i)}(t), t), \quad S(0) = \hat{S}_{\text{lat}}(0) = \text{Encoder}(S^{\text{past}}, X^{\text{past}}). \tag{1}$$

In practice, the solution of equation 1 can be approximated by a forward-Euler style discretization:

$$s_i^{t_{h+1}} = s_i^{t_h} + (t_{h+1} - t_h)\text{NN}_i(S_{\text{pa}(i)}^{t_h}, X_{\text{pa}(i)}^{t_h}, t_h), \tag{2}$$

where $h = 0, 1, \ldots$, and we switched the notation from $S_{\text{pa}(i)}(t)$ to $S_{\text{pa}(i)}^t$ to emphasize the transition from continuous time-domain to a discrete time grid. In our implementation, we choose the encoder layer to be a standard LSTM and the feed-forward neural networks $\{\text{NN}_i\}_{i=1}^n$ to be standard MLPs.

**2.4 Reduction Algorithm: Hybrid graph sparsification (HGS)**

**Step 1: merging maximal strongly connected components** Given a directed graph representation of a mechanistic ODE system $G = (E, V)$ (since the dependency on the mechanistic ODE system is clear from context, we will omit the $M$ subscript to simplify notations), we first collapse all maximal strongly connected components (MSCCs) in $G$ into super-nodes to make it an RDAG (note that we allow self loops). This is implemented by first partitioning $V$ into disjoint subsets of MSCCs $C_i$:

$$V = \sqcup_{i=1}^k C_i, \quad \forall i \neq j, \ C_i \cap C_j = \emptyset.$$

Next, we define a super-node set $V^a$ by mapping each MSCC in $V$ to a super-node in $V^a$ :

$$V^a = \{c_i^a \mid C_i \subseteq V, \ 1 \leq i \leq k\}.$$

Then, to define edges between super-nodes, for each directed edge $(u, v) \in E$, we add $(c_i^a, c_j^a)$ to the super-edge set $E^a$, where $C_i$ and $C_j$ are the two (not necessarily different) MSCCs contains $u$ and $v$, respectively:

$$E^a = \{(c_i^a, c_j^a) \mid (u, v) \in E, \ u \in C_i, v \in C_j\}.$$

We denote the resulting super-graph as $G^a = (V^a, E^a)$. Each super-node in $V^a$ may collapse multiple observable state nodes into a single "super-state" node, whose feature is defined as the concatenation of all observable node features within its MSCC. Let $S_{\text{obs}}^a \subset V^a$ be the set of super-nodes whose MSCCs contain at least one observable state node, and $X^a \subset V^a$ the set corresponding to $X \subset V$ in $G$. For consistency, we similarly define $S^a, S_{\text{lat}}^a, S_{\text{pa}(i)}^a$, and $X_{\text{pa}(i)}^a$. This yields an RDAG representation of the mechanistic model, $G^a$.

**Rationale of step 1** Transforming the original graph into an RDAG via collapsing the MSCCs reveals high-level causal structure of the system, simplifies the interpretation, and provides a topological ordering with acyclic structure that improves training stability, as feedback loops are known to cause blow-ups, exploding gradients and stiffness of the ODE system. With cycles, many complicated constraints on the system parameters are needed to simultaneously control stiffness, blow-ups and exploding gradients. Without cycles, the system Jacobian is upper triangular after proper rearrangement, and the corresponding eigenvalues are simply its' diagonal elements, allowing substantially fewer and simpler constraints on parameters to ensure system stability. (see Appendix A8 for detailed discussions and examples). In addition, replacing each MSCC with a self-loop does not sacrifice much predictive power because neural networks are capable of approximating the effect of complex intra-component dynamics—a key motivation of hybrid modeling Raissi et al. (2019).

**Step 1 Customization:** While we have chosen the default set-up of HGS step 1 to collapse all MSCCs, users may, based on application needs and their own domain knowledge, choose not to collapse certain MSCCs, and Step 2 and 3 of HGS will still be compatible in these cases. Causal interpretability can be preserved via temporal unfolding, where feedback loops are resolved into time-lagged dependencies between distinct temporal instances of the state variables.

**Step 2: augmenting graph with simpler shortcuts** Next, we identify key mechanistic pathways and augment them with simpler shortcuts for potential model reduction. To this end, let $D_{x,s}$ be the set of nodes, whose removal disconnects $x$ and $s$ in $G^a$:

$$D_{x,s} = \{v \in V^a \mid v \neq x, v \neq s, \ s \text{ no longer reachable from } x \text{ in } G^a \text{ after removing } v\},$$

and let $G_{x,s}^a$ be the sub-graph induced by $\{x, s\} \cup D_{x,s}$:

$$G_{x,s}^a = (V_{x,s}^a, E_{x,s}^a), \quad V_{x,s}^a = \{x, s\} \cup D_{x,s}, \quad E_{x,s}^a = \{(u, v) \in E^a \mid u, v \in V_{x,s}^a\}.$$

Define the partial transitive closure $G_{x,s}^{a,c}$ of $G_{x,s}^a$ to be:

$$G_{x,s}^{a,c} = (V_{x,s}^a, E_{x,s}^{a,c}), \quad E_{x,s}^{a,c} = \begin{cases} \overline{E_{x,s}^a} \setminus \{(x,x)\} & (x, s) \in E_{x,s}^a, \\ \overline{E_{x,s}^a} \setminus \{(x,x), (x,s)\} & (x, s) \notin E_{x,s}^a, \end{cases}$$

where $\overline{E_{x,s}^a}$ is the edge set of the transitive closure of $G_{x,s}^a$ using the reachability relation of $G^a$. Finally, we augment the original RDAG $G^a$ with the additional edges from partial transitive closures of pathway sub-graphs to form the augmented RDAG $G^{a,c}$:

$$G^{a,c} = (V^a, E^{a,c}), \quad E^{a,c} = E^a \cup_{x \in X^a, s \in S_{\text{obs}}^a} E_{x,s}^{a,c}.$$

and $G^{a,c}$ will be the graph used for step 3. Appendix A2.1 shows an example of $G$ vs $G^{a,c}$.

**Intuition and rationale of step 2** Intuitively, one may think of a physiological path as a student's high-school journey moving through grades 9 to 12. Normally, the student progresses step by step—9 $\rightarrow$ 10 $\rightarrow$ 11 $\rightarrow$ 12. A transitive closure adds all possible "skip-grade" links, letting the student jump directly from grade 9 to 11 or 12, or from 10 to 12, as long as they always move to a higher grade (obey the reachability relations). A partial transitive closure is a more cautious version: it allows some skipping but forbids overly aggressive jumps, like going straight from grade 9 to 12. The idea is that, just as students progress at different speeds, biological processes/pathways in physiological systems also vary in how many intermediate states they pass through and can therefore often be better modeled with fewer latent states. For example, quasi-steady-state approximations in chemical kinetics eliminate fast variables by assuming equilibrium. By adding shortcuts (transitive closure), the model gains flexibility to capture these differences without discarding realistic reachability constraints.

**Step 2 Customization** Using a partial (rather than full) transitive closure is a choice made by the authors in the context of glucose modeling because it prevents introducing direct input–output edges unsupported by the mechanistic model, and preserving some latent dynamics has been shown to be important (Dalla Man et al., 2009). Similar to step 1, rather than following this default set-up, users may choose to include full transitive closure or omit selected short-cut paths based on their needs.

**Step 3: applying a mix of $L_1$ and $L_2$ regularization** Given a processed RDAG $G^{a,c}$, to automatically remove redundant edges and nodes, a natural way is to associate a weight with each edge and apply $L_1$ penalty to shrink weights of redundant edges to zero in the style of LASSO regularized regression. In the context of MNODE, a straight-forward formulation would be to modify Equation 2 to:

$$\frac{ds_i^a(t)}{dt} = \text{NN}_i(W \odot S_{\text{pa}(i)}^a(t), W \odot X_{\text{pa}(i)}^a(t), t; \Theta_i),$$

where the $i$th neural network is parametrized by $\Theta_i$, $W = \{w_{(u,v)} \mid (u,v) \in E^{a,c}\}$ is the set of edge-specific weights, and $\odot$ stands for element-wise multiplications:

$$W \odot S_{\text{pa}(i)}^a(t) = \{w_{(s,s_i)} \cdot s(t) \mid s \in S_{\text{pa}(i)}^a\}, \quad W \odot X_{\text{pa}(i)}^a(t) = \{w_{(x,s_i)} \cdot x(t) \mid x \in X_{\text{pa}(i)}^a\}.$$

Given the mechanistic RDAG $G^{a,c}$ defining the MNODE structure, NN parameter $\Theta = (\Theta_1, \ldots, \Theta_n)$, and edge weights $W$, one may predict the state variable values (i.e. node features) at $t_1, \ldots, t_q$ with an initial condition estimate $\hat{S}^{a,t_0}$ and future exogenous inputs $X^{a,\text{F}}$ over time. These predictions can be recursively calculated based on (2):

$$\hat{s}_i^{a,t_{h+1}} = \hat{s}_i^{a,t_h} + (t_{h+1} - t_h)\text{NN}_i(W \odot \hat{S}_{\text{pa}(i)}^{a,t_h}, W \odot X_{\text{pa}(i)}^{a,t_h}, t_h; \Theta_i), \quad \text{with} \quad \hat{s}_i^{a,t_0} = s_i^{a,t_0}.$$

We denote the resulting prediction of observable states $\hat{S}_{\text{obs}}^a$ at $t_h$ by $\hat{S}_{\text{obs}}^{a,t_h}(S^{a,t_0}, X^{a,\text{F}}; \Theta, W, G^{a,c})$ to emphasize its dependence on relevant model parameters, initial condition and exogenous input. We estimate encoder and decoder parameters simultaneously by minimizing the mean-squared-error loss function. To encourage graph sparsity while retaining identifiability, we also place a combination of $L_1$ and $L_2$ regularization on edge weights $W$ and model weights $\Theta$ to form the final loss function:

$$\sum_{\text{cases},h} \left\| S_{\text{obs}}^{a,t_h} - \hat{S}_{\text{obs}}^{a,t_h}\left(\hat{S}^{a,t_0}(D^{a,\text{P}};\beta), X^{a,\text{F}}; \Theta, W, G^{a,c}\right) \right\|_2^2 + \lambda_1 \sum_{(u,v) \in E^{a,c}} |w_{u,v}| + \lambda_2 \|\Theta\|_2^2. \quad (3)$$

where $D^{a,\text{P}} = (S_{\text{obs}}^{a,\text{P}}, X^{a,\text{P}})$ are observed data available at time 0, $\hat{S}^{a,t_0}(\cdot; \beta)$ is the encoder generating the initial condition of the system, and $\lambda$ is a penalty parameter. The $L_1$ penalty on edge weight $W$ is designed to encourage sparsity and the $L_2$ penalty on decoder parameters is to boost identifiablility.

**Equivalence to first-layer group LASSO:** The above regularization is closely related to the idea of first-layer group LASSO mentioned in (Cherepanova et al., 2023). It can be shown (see Appendix A2.2) that Equation 3 is equivalent to

$$\sum_{\text{cases},h} \left\| S_{\text{obs}}^{a,t_h} - \hat{S}_{\text{obs}}^{a,t_h}\left(\hat{S}^{a,t_0}, X^{a,\text{F}}; (\Gamma, \tilde{\Theta}), \mathbf{1}, G^{a,c}\right) \right\|_2^2 + \lambda_2 \|\tilde{\Theta}\|_2^2 + \lambda_3 \sum_{(u,v) \in E^{a,c}} \|\Gamma_{(v,u)}\|_2^{2/3},$$

where $\lambda_3 = 3 \times 2^{-2/3} \lambda_1^{2/3} \lambda_2^{1/3}$, $\tilde{\Theta}$ represent all non-first-layer weights in the MLPs, and $\Gamma_{v,u} = w_{(u,v)} \Theta_{(u,v)}$ is the $w_{(u,v)}-$scaled vector consisting of first-layer-multiplication weights associated with the edge $(u, v)$. The $\sum_{(u,v) \in E^{a,c}} \|\Gamma_{(v,u)}\|_2^{2/3}$ term is a variant of standard group LASSO penalty encouraging group sparsity, and the vector $\Gamma_{(u,v)} = 0 \iff$ removing edge $(u, v) \in E^{a,c}$. Compared to the standard group LASSO penalty that raises $\|\Gamma_{(v,v)}\|_2$ to power of 1, a smaller exponent encourages stronger group sparsity with steeper gradient towards 0. Operationally, the regularization parameters $\lambda$ are selected via $K$-fold cross-validation (CV).

**Step 3 Customization** In step 3, while the default set-up is to penalize all edge weights $W$, if certain edges are deemed indispensable or more important by the user, they can be taken out of the regularization term or be given separately a customized $\lambda$.

**2.5 Important note: implausibility of true support recovery** It should be pointed out that our method is not meant for true support recovery (i.e. recover the true underlying causal graph of the data generating process) because the expressivity of neural networks lead to equivalent MNODE models whose underlying graphs are different (Xia et al., 2021), and the task of recovering the true graph structure is therefore theoretically implausible without making strong assumptions about ground truth. **Our goal is to efficiently induce model sparsity and facilitate the generation of data-driven hypotheses, which require further clinical validation, without making strong assumptions about the ground truth other than that it is more sparse than the original mechanistic prior.**

# 3 RELATED WORK

**Sparsity, LASSO, and Generalization** LASSO-style penalties have long been used to induce sparsity. Applications range from residual networks (Lemhadri et al., 2021), varying-coefficient models (Thompson et al., 2023), and CNNs via group LASSO (Liu et al., 2015; Wen et al., 2016), to feature selection in MLPs (Zhao et al., 2015; Sun et al., 2016; Wang et al., 2017) and local linear sparsity (Ross et al., 2017). However, these methods often yield sparsity patterns that are hard to interpret. Our approach extends LASSO to MNODE while grounding the learned sparsity in mechanistic graph structures, providing interpretability and domain-aligned insights. Prior work has shown that reduced neural networks can generalize as well as, or even better than, their unreduced counterparts (Gale et al., 2019; Bartoldson et al., 2020; Hoefler et al., 2021). For graph-based model reduction, You et al. (2020) found that mildly sparse relational graphs often outperform fully connected ones. Our results support this: MNODEs built from lightly pruned graphs perform better than those using dense graphs in low-data regimes.

**Graph Sparsification via Optimization** Our method is related to optimization-based graph sparsification approaches for GNNs. For example, Li et al. (2020) constrained the $L_0$ norm of the adjacency matrix, Jiang et al. (2021) used elastic net penalties, and Jiang et al. (2023) applied exclusive group LASSO to encourage neighborhood sparsity. Unlike these methods, we do not sparsify the adjacency matrix directly. Instead, we sparsify edge weights in message passing, reducing influence from less informative neighbors. Crucially, GNN sparsification methods are typically data-driven and ignore mechanistic structure. In contrast, our method is structure-aware: it begins with a domain-informed pruning step that restricts the search to physically plausible graphs. This makes our approach more suitable for hybrid models like the Graph Network Simulator (Sanchez-Gonzalez et al., 2020) that require mechanistic consistency. Finally, while Zou et al. (2024) used a greedy, stepwise reduction scheme, our method is more computationally efficient and yields better performance—analogous to the gains of LASSO over stepwise selection in linear models (Hastie et al., 2020).

**Landscape of Hybrid Modeling and Trade-offs** There are many variants of MNODE in the current landscape of hybrid modeling with different degrees of hybridization and strength of mechanistic

prior. In Zou et al. (2024), a more general form of mechanistic neural ODE is proposed as:

$$\frac{dS(t)}{dt} = c_1 f_m(S(t), X(t), t; \beta(t)) + (1 - c_1) f_{nn1}(S(t), X(t), c_2 Z(t); c_3 A_m + (1 - c_3)\mathbf{1})$$

$$\beta(t) = c_4 \beta_m + (1 - c_4) f_{nn2}(S(t), X(t), t), \text{ and } \frac{dZ(t)}{dt} = f_{nn3}(S(t), X(t), Z(t), t),$$

where $f_m, \beta_m$ and $A_m$ represent the functional form, model parameters and dependency structure (adjacency matrix) of the mechanistic prior, respectively, and $f_{nni}, i = 1, 2$ are neural networks. For $c_1 = c_2 = 0, c_3 = 1$, one recovers MNODE. The deep mechanistic simulator in Miller et al. (2020) uses $c_1 = 1, c_4 = 0$. The neural closure learning model in Gupta & Lermusiaux (2021) uses $c_1 = 0.5, c_2 = c_3 = c_4 = 0$. The hybrid ODE model in Qian et al. (2021) uses $c_1 = 0.5, c_2 = 1, c_3 = 0$, while the standard black-box neural ODE uses $c_1 = c_3 = 0, c_2 = 1$. In general, the more mechanistic components, the more constraints on the function space of the resulting hybrid model. In data-limited regimes, such constraints help the model quickly learn reasonable representations but also undermine the model's ability to capture complex real-world dynamics not represented by the mechanistic prior. On the other hand, the less mechanistic prior, the more flexible the hybrid model becomes. While the model gains more freedom to learn arbitrary patterns, it is also exposed to risks of overfitting and high variance. The challenge is to find the sweet spot: using just enough mechanistic guidance to keep the model grounded without too much constraining.

## 4 EXPERIMENTS

### 4.1 Experiments on synthetic data

**General Setting** We assume the data are generated by an unknown sparse dynamical system (ODE), but the mechanistic model is overly complex and contains redundant variables. We show that HGS produces models that are more sparse, predictive and robust than those obtained by existing methods.

**Data generation** We consider two sparsity regimes: true sparsity–redundant feature have zero effect size, and quasi sparsity–redundant features have non-zero but small effect sizes. Our synthetic data are generated from the following two controlled ODE systems respectively:

$$\text{True Sparsity:} \quad \frac{ds_1(t)}{dt} = 0.5[s_1(t) - 1] + 4x_1(t),$$

$$\text{Quasi Sparsity:} \quad \frac{ds_1(t)}{dt} = 0.5[s_1(t) - 1] + 4\sum_{j=1}^{|X|} \frac{x_j(t)}{10^{j-1}}.$$

We generated time series samples of length $q = 60$ based on the forward Euler numerical integration scheme with time step $\nabla t = 0.05$ over the domain $t \in [0, 0.3]$ with zero initial conditions.

**Training Sample Size** To study the effect of sample size on our method and validate its effectiveness in limited data regime, for each sparsity regime, we generated 40 independent training sets of size 100 and 1000 respectively, as well as a held-out test set of size 10,000.

**Starting graph** Assuming the true data generating process is unknown, we consider two settings in which the starting "mechanistic" graph contains redundant structures: (1) a refined graph whose redundant part contains 3 input nodes, 1 latent node, and 1 latent cycle; (2) a comprehensive graph whose redundant part contains 6 input nodes, 3 latent nodes and 3 latent cycles (See Appendix A4.1 for illustrations). All redundant input variables are generated from independent $\mathcal{N}(0, 0.5)$.

**Baseline models** We selected the following baseline models: (1) Block-box sequence models including: LSTM, Black-box neural ODE (BNODE), temporal convolution network (TCN) (Lea et al., 2016), Diagonal S4 (S4D) (Gu et al., 2022) and vanilla transformer (Trans); (2) MNODE reduced by other reduction methods including: no reduction (NR), NeuralSparse (NS) (Zheng et al.,

2020), exclusive group LASSO (EGL) (Jiang et al., 2023), elastic net (EN) (Jiang et al., 2021), random search (RD) and step-wise greedy search (GD). Implementation details are in Appendix A4.

**Evaluation and metrics** All evaluations are performed on the test set (see Appendix A5). We report RMSE with 1-sigma standard error (SE) for predictive performance, Peak (worst-case) RMSE for robustness and Effective Number of Parameters (ENP, average number of parameters whose magnitude is $> 10^{-3}$ under CV-selected hyperparameter setting) for model reduction. Additional metrics including MAPE, Peak MAPE and correlation are reported in A7.1.

**Results** As shown in Figure 1, HGS outperforms black-box models at small sample sizes, with the gap narrowing as sample size increases. At $n = 1000$, TCN surpasses HGS in RMSE, but HGS retains superior robustness. This reflects a known trade-off: with more data, the bias from regularization may outweigh its variance reduction benefits, though the latter still improves worst-case behavior. Compared to other reduction methods, HGS consistently achieves the best performance. However, when the input graph is already refined, the gains are modest—as expected, since most reasonable methods perform well in this setting. On graphs with substantial redundancy, HGS's advantage becomes both large and statistically significant. In such cases, regularization alone struggles to recover signal, especially under quasi-sparsity. HGS's hybrid pruning, when paired with regularization, addresses this challenge effectively (See Appendix A7.2 for an ablation study). It also yields the fewest effective nonzero parameters (ENPs), highlighting its strength in inducing sparsity.

### 4.2 Experiments on real-world data: blood glucose forecasting for T1D patients

**Introduction and background** For the real-world data experiment, we focus on modeling the carbohydrate-insulin-glucose dynamics in patients with Type 1 diabetes (T1D). T1D patients have impaired insulin production and therefore rely on constant external insulin delivery to regulate their blood glucose level, making their glycemic regulation a challenging dynamical system to model, especially during periods of physical activities. In this case, the latent states represent different physiological compartments within the human body and the observed state is blood glucose level. We choose our mechanistic model to the 2013 Version of the UVA-Padova model (Man et al., 2014) (Appendix A6), which is FDA-approved for modeling glycemic response in T1D patients.

**Data** Our data are from the T1D Exercise Initiative (T1DEXI) (Riddell et al., 2023), which is available to public at `https://doi.org/10.25934/PR00008428` (Jaeb Center for Health Research Foundation, 2022). After pre-processing (Appendix A3), the final data contain 342 time series from 105 patients. Each time series consists of 54 measurements, taken 5 minutes apart, of blood glucose level and exogenous inputs including carbohydrate intake, insulin injection, heart rate and step count from 210 minutes before to 60 minutes after the onset of an exercise instance. We set the first 210 minutes as history window and the remaining 60 minutes as the prediction window.

**Baseline models** We also consider MNODE reduced by domain knowledge (DK) (Zou et al., 2024).

**Evaluation and metrics** We adopt repeated CV (see Appendix A5) to estimate various metrics of interest. Specifically, we split data randomly at the exercise instance level to evaluate the average predictive performance on both intra- and inter-patient instances, reflecting the likely real-world deployment of the prediction algorithm to both existing and new patients. In addition, since intra-patient variability can be as large as inter-patient variability in T1D modeling (Moscoso-Vasquez et al., 2016; Bell et al., 2021; Laguna et al., 2014), we anticipate similar results from patient-level cross-validation (see Appendix A4.2 for details and empirical evidence). In addition to all 6 metrics used in synthetic set-ups, we include model variance for measuring robustness and an clinical significance metric, Diagnostic Accuracy–accuracy of classifying a patient as hyper ($\geq$ 180 mg/dl), in-range (80-180 mg/dl) or hypo ($\leq$ 80 mg/dl) using model predictions. All metrics, except Peak RMSE, Peak MAPE and ENP, are reported with 1-sigma SE.

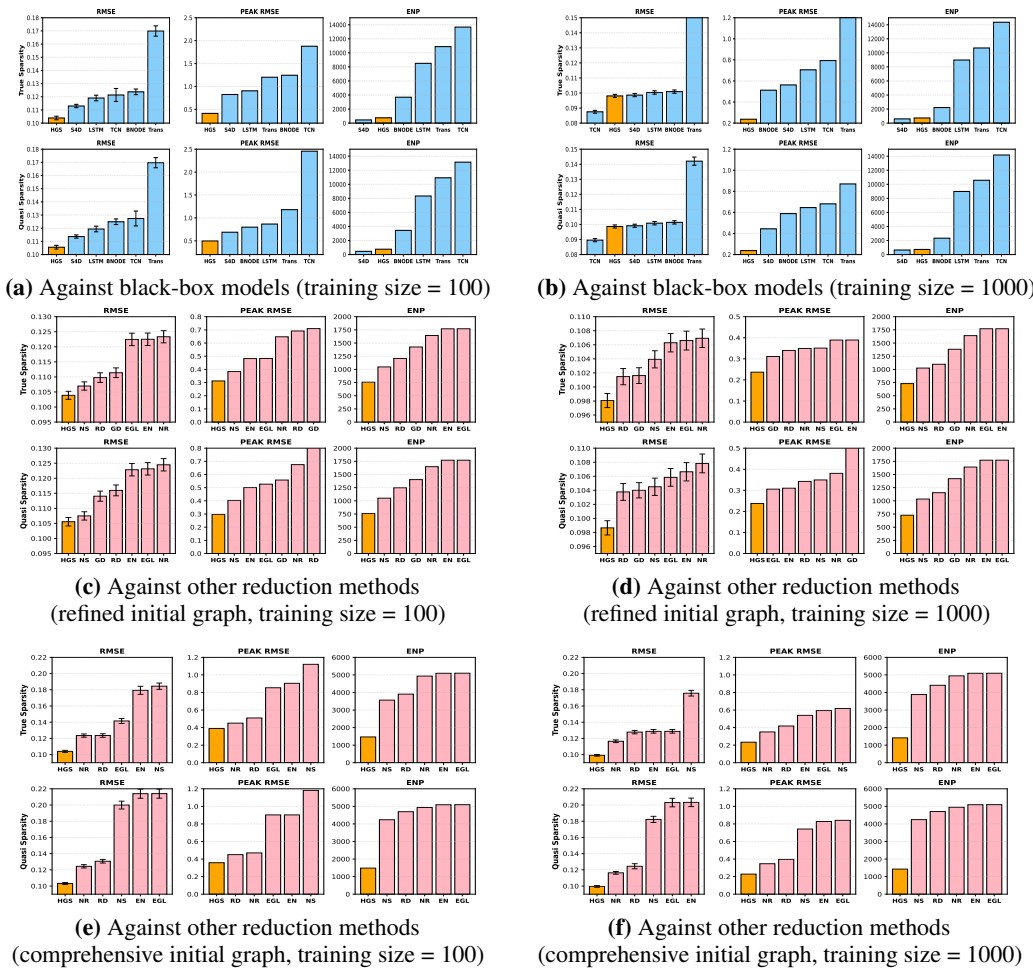

Figure 1: Comparison across different evaluation settings. GD is omitted for comprehensive initial graph as training takes an unreasonable amount of time. Models are sorted from best to worst.

**Results** As shown in Figure 2(a), MNODE_HGS significantly outperforms traditional black-box models across all metrics, while using fewer parameters. This highlights the benefits of incorporating mechanistic structure and graph sparsification into model design. Figure 2(b) compares HGS with alternative reduction methods applied to MNODE. HGS consistently yields better predictive performance, particularly in peak RMSE, suggesting greater robustness. To visualize the learned structures, we plot in Figure 2(c) edge-weighted adjacency matrices of the reduced graphs, averaged over 10 runs. Unlike other methods, HGS not only promotes sparsity but also introduces new structural shortcuts that are otherwise inaccessible to regularization-based approaches.

**Ablation study on model components** To assess the contribution of each design step in Section 2, we conduct an ablation study using models trained with various subsets of the proposed pipeline. As shown in Figure 2(d), removing any single step leads to a marked drop in performance, underscoring the importance of all three components for the success of MNODE_HGS.

**Mechanistic interpretation of results** HGS can yield interpretable and biologically plausible new insights. For example, HGS chooses to eliminate edges corresponding to glucagon feedback loops, which suggests that impaired glucagon response during hypoglycemia (Seaquist et al., 2013) may also persist during exercise-induced hypoglycemia—a novel hypothesis that could guide future investigations.

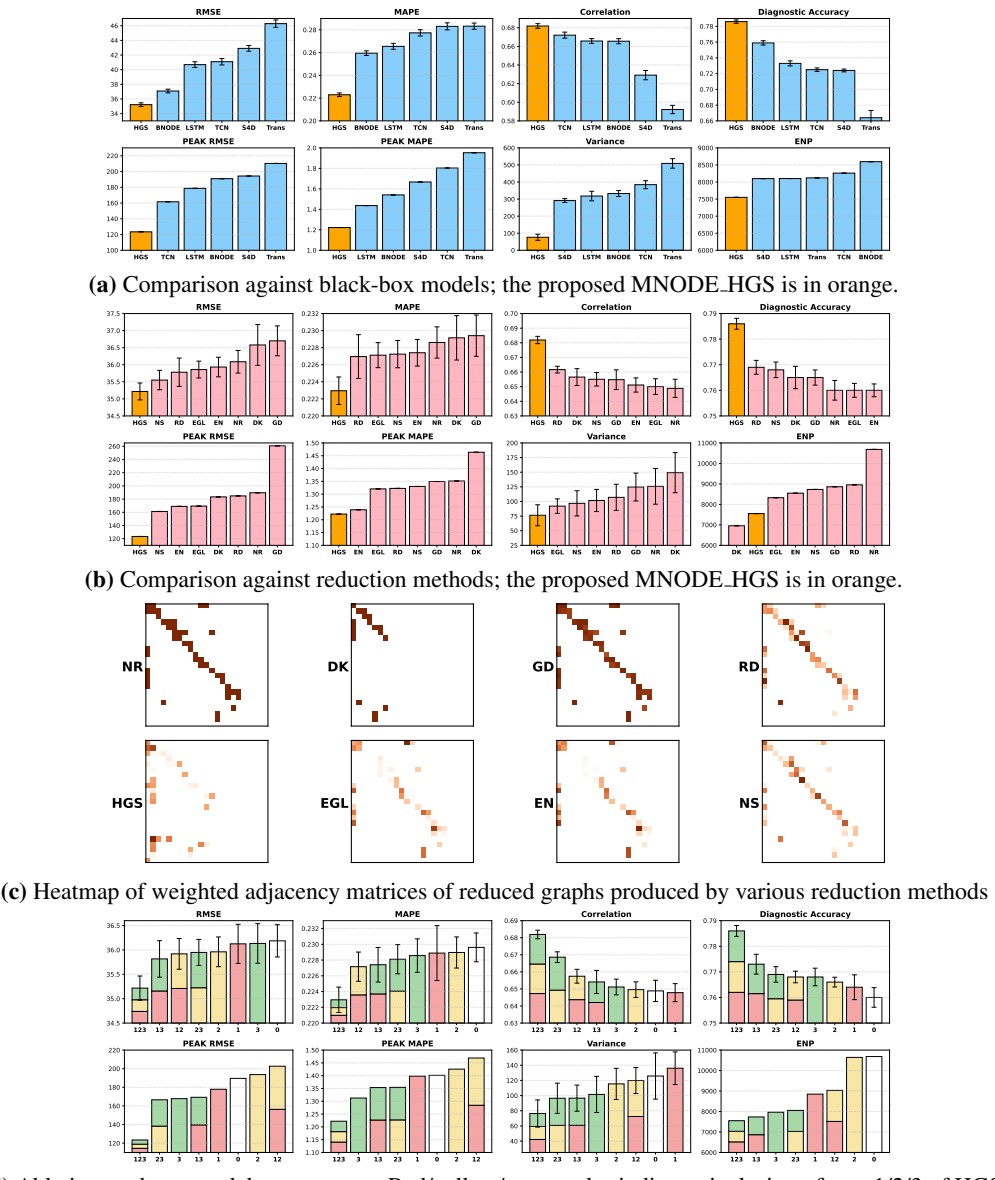

(a) Comparison against black-box models; the proposed MNODE_HGS is in orange.

(b) Comparison against reduction methods; the proposed MNODE_HGS is in orange.

(c) Heatmap of weighted adjacency matrices of reduced graphs produced by various reduction methods

(d) Ablation study on model components. Red/yellow/green color indicates inclusion of step 1/2/3 of HGS.

Figure 2: Combined Results for Real-world Experiments. Models are sorted from best to worst.

# 5 BROADER IMPACT

We propose a three-step procedure to simplify the mechanistic graph underlying MNODEs, aiming to improve prediction, robustness and interpretability. This is especially useful in biomedical domains, where models often involve complex biological processes and high-quality data are limited. By leveraging domain-informed graph refinement, structural pruning, and edge-weight sparsification, our method produces compact and predictive models that align with mechanistic priors while reducing overfitting. This enables more transparent and data-efficient modeling, potentially accelerating discovery in systems biology, personalized medicine, and related fields. More broadly, our framework contributes to the effort to integrate domain knowledge into deep learning in a principled way. It provides insights into which components (e.g. cycles, delays) of a mechanistic model are predictive, offering guidance for experimental focus and hypothesis generation.

## ACKNOWLEDGMENT

This publication is based on research using data from Jaeb Center for Health Research Foundation that has been made available through Vivli, Inc. Vivli has not contributed to or approved, and is not in any way responsible for, the contents of this publication. We thank Alex Wang, Dessi Zeharieva, Emily Fox, Matthew Levine and Ramesh Johari for providing assistance with data acquisition and preliminary discussions on T1D and model reduction.

## REPRODUCIBILITY STATEMENT

Our data processing pipeline is described in Appendix A3. Model implementation and training details are provided in Appendix A4.2. Model evaluation details are provided in Appendix A5. The mechanistic model used is described in Appendix A6. Our code is also submitted as supplementary materials.

## LLM USAGE

LLM is only used to aid and polish the writing of the paper. We did not use LLM for any other purpose.

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

## A  APPENDIX

INDEX OF APPENDIX CONTENTS

## A2  ADDITIONAL DETAILS ABOUT HGS

### A2.1  ILLUSTRATION OF HGS

Figure 3 shows an illustrative example of step 1 and 2 of HGS for a simple graph.

### A2.2  PROOF FOR GROUP LASSO EQUIVALENCE

To make the connection, note that in the context of MLP model computation, applying a scaling parameter $w$ to an input feature is equivalent to re-scaling all the first-layer weights linked to the corresponding feature by a factor of $w$. Therefore, suppose we let $\Theta_{(u,v)}$ be the vector consisting of first-layer-multiplication weights associated with the edge $(u,v) \in E^{a,c}$, and $\tilde{\Theta}$ be the vector consisting of all non-first-layer-multiplication model parameters in all MLPs, we can rewrite $\hat{S}_{\text{obs}}^{a,t_h}(\cdot,\cdot;\Theta,W,G^{a,c})$ and the regularization term in (3) as

$$\hat{S}_{\text{obs}}^{a,t_h}\left(\cdot,\cdot;\left(\{w_{(u,v)}\Theta_{(u,v)}|(u,v)\in E^{a,c}\},\tilde{\Theta}\right),\mathbf{1},G^{a,c}\right)$$

$$\text{and} \sum_{(u,v)\in E^{a,c}}\left(\lambda_1|w_{(u,v)}|+\lambda_2\|\Theta_{(u,v)}\|_2^2\right)+\lambda_2\|\tilde{\Theta}\|_2^2,$$

respectively, where $\mathbf{1}$ is the set of unit edge weights corresponding to the unweighted specification. Thus, letting $\Gamma_{v,u}=w_{(u,v)}\Theta_{(u,v)}$ and $\Gamma=\{\Gamma_{(u,v)}|(u,v)\in E^{a,c}\}$, the loss function (3) can be reparametrized as:

$$\sum_{\text{cases},h}\left\|S_{\text{obs}}^{a,t_h}-\hat{S}_{\text{obs}}^{a,t_h}\left(\hat{S}^{a,t_0}(D^{a,\text{P}};\beta),X^{a,\text{F}};(\Gamma,\tilde{\Theta}),\mathbf{1},G^{a,c}\right)\right\|_2^2$$

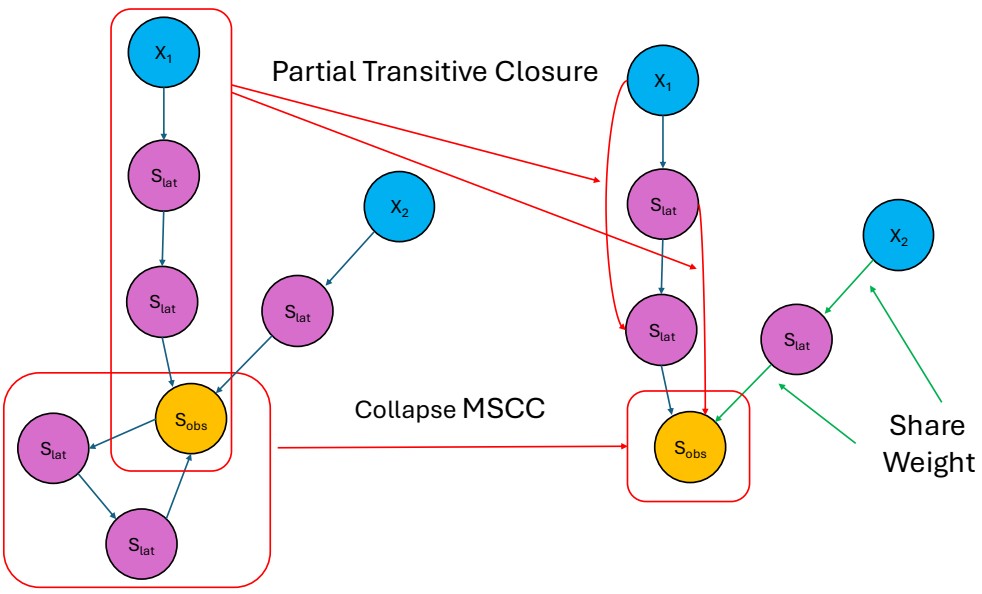

Figure 3: Step 1 and 2 of the Hybrid Graph Sparsification Algorithm

$$+\lambda_2\|\tilde{\Theta}\|_2^2 + \sum_{(u,v)\in E^{a,c}} \left(\lambda_1|w_{(u,v)}| + \lambda_2\frac{\|\Gamma_{(u,v)}\|_2^2}{w_{(u,v)}^2}\right). \tag{4}$$

Note that $\{w_{(u,v)} \mid (u,v) \in E^{a,c}\}$ only appears in the last term of (4), One may minimize this loss with respect to $W$, while fixing all other parameters in 4, and obtain the minimizer

$$w^*_{(u,v)} = \left(\frac{2\lambda_2\|\Gamma_{(u,v)}\|_2^2}{\lambda_1}\right)^{\frac{1}{3}}.$$

Substituting $w^*_{(u,v)}$ back into (4) gives the desired loss function with respect to only $\Gamma, \tilde{\Theta}$ and $\beta$ :

$$\sum_{\text{cases},h} \left\|S^{a,t_h}_{\text{obs}} - \hat{S}^{a,t_h}_{\text{obs}}\left(\hat{S}^{a,t_0}, X^{a,\text{F}};(\Gamma,\tilde{\Theta}),\mathbf{1},G^{a,c}\right)\right\|_2^2 + \lambda_2\|\tilde{\Theta}\|_2^2 + \lambda_3 \sum_{(u,v)\in E^{a,c}} \|\Gamma_{(v,u)}\|_2^{2/3}.$$

If we replace $|w_{(u,v)}|$ in (3) by $w^2_{(u,v)}$, then the same derivation can show that the equivalent loss function becomes

$$\sum_{\text{patients},h} \left\|S^{t_h}_{\text{obs}} - \hat{S}^{t_h}_{\text{obs}}\left(\hat{S}^0(D^{\text{P}};\beta), X^{\text{F}};(\Gamma,\tilde{\Theta}),\mathbf{1},G^{a,c}_M\right)\right\|_2^2$$

$$+\lambda_2\|\tilde{\Theta}\|_2^2 + 2\sqrt{\lambda_1\lambda_2} \sum_{(u,v)\in E^{a,c}_M} \|\Gamma_{(v,u)}\|_2$$

with a standard group LASSO penalty on $\Gamma$.

## A3 APPENDIX: REAL-WORLD DATA PRE-PROCESSING

### A3.1 SELECTION

We select patients on open-loop pumps with age under 40 and body mass index (BMI) less than 30. From exercise instances recorded by these patients, we focus on the time window between 210 minutes before exercise onset and 60 minutes after exercise onset, and select those that satisfy the following conditions:

1. The exercise lasted for at least 30 minutes.

2. There are no missing blood glucose readings (recorded every 5 minutes) or heart rate reading (recorded every 10 seconds) in the time window of interest.

3. There is at least one carbohydrate intake in the time window of interest.

With the above selection criteria, we end up with 324 exercise instances from 105 patients.

## A3.2 Features, Units and Interpolation

For each selected exercise instance, we use the following features derived from the T1DEXI raw data:

| Feature | File | Data Field | Unit | UVASim Unit |
|---|---|---|---|---|
| CGM Reading (Blood Glucose Concentration) | FA | FATEST | mg/dL | mg/dL |
| Basal Insulin Rate | FACM | FATEST | U/hour | U/min |
| Bolus Insulin | FACM | FATEST | U | U/min |
| Dietary Total Carbohydrate | FAMLPM | FATEST | g | mg/min |
| Verily Heart Rate | VS | VSCAT | bmp | NA |
| Verily Step Count | FA | FATEST | count | NA |

Table 1: Summary of features and their units in T1DEXI and UVA/Padova

| Feature | Notation | Ascending time |
|---|---|---|
| $i$-th entry of CGM reading and its time-stamp | $(g_i, t_i^g)$ | Yes |
| $i$-th entry of basal insulin rate and its time-stamp | $(a_i, t_i^a)$ | Yes |
| $i$-th entry of bolus insulin and its time-stamp | $(b_i, t_i^b)$ | Yes |
| $i$-th entry of carb and its time-stamp | $(m_i, t_i^m)$ | No in original Data, need sorting |
| $i$-th entry of Verily HR and its time-stamp | $(h_i, t_i^h)$ | Yes |
| $i$-th entry of Verily Step Count and its time-stamp | $(v_i, t_i^v)$ | Yes |

Table 2: Notations for various features

## A3.3 Interpolating Basal Flow Rate

Since the basal flow rate is already RATE, interpolating it is relatively straightforward. We approximate basal flow rate with a step function where the magnitude and time of jumps are determined by $a_i$ and $t_i^a$ respectively:

$$f_a(t) = \begin{cases} a_i/60 & t_i^a \le t < t_{i+1}^a, \quad 1 \le i < N \\ a_N/60 & t_N^a \le t \\ 0 & \text{otherwise} \end{cases},$$

where the unit for $f_a$ is U/min, the same as the units used for insulin delivery rate in UVA/Padova; and $N$ is the total number of basal insulin rate data entries.

## A3.4 Interpolating Bolus Insulin Rate

The recorded data are dosage and the corresponding time. To convert dosage to rate, we refer to real-world experiences where most open loop pumps inject a dose of bolus insulin at a rate of 1.5U/min. We ignore basal insulin when computing the effective bolus rates because basal insulin rates contribute negligibly to the maximum rate

First, to account for cases of overlapping bolus doses (a new dose is applied while the previous dose has not been completely delivered), we pre-process the bolus insulin data with the following trick: if two bolus doses $(b_j, t_j^b), (b_{j+1}, t_{j+1}^b)$ overlap (i.e. $t_{j+1}^b < t_j^b + b_j/1.5\text{min}$), it is as if we only introduced one dose of bolus insulin of $b_j + b_{j+1}$ U at time $t_j^b$. We can recursively apply this trick to combine all overlapping doses (see Algorithm 1 for details), after which we end up with a new list bolus insulin data $(\hat{b}_j, \hat{t}_j^b)$ in which we can assume there are no overlapping doses.

---

**Algorithm 1** Pre-process Bolus:

---

**Input:** Bolus Insulin Data $B = \{(b_j, t_j^b)\}_{j=1}^N$ where $b_j$ is in U and $t_b^j$ is in min.
$i = 1$
Initialize $\hat{B}$ as an empty list
**while** $i <= \text{length}(B)$ **do**
   **if** $t_{i+1}^b < +t_i^b + b_i/1.5$ **then**
      $B(i) = (b_i + b_{i+1}, t_i^b)$
      delete $(b_{i+1}, t_{i+1}^b)$ from B in place
   **else**
      $i = i + 1$
      Append $B(i)$ to $\hat{B}$
   **end if**
**end while**
**Return** $\hat{B}$

---

Next, we define the continuous-time bolus delivery function $f_b(t)$ as :

$$f_b(t) = \begin{cases} 1.5 & \hat{t}_j^b \leq t < \hat{t}_j^b + (b_j/1.5)\text{min} \\ 0 & \text{otherwise} \end{cases}$$

where the unit for $f_b(t)$ is U/min as well.

### A3.5 Interpolating Carbohydrate Intake Rate

Suppose the patient consumed carbohydrates at a given time in the T1DEXI dataset. We assume a constant meal consumption rate of 45000 milligrams per minute:

$$f_m(t) = |M|45000, \quad M = \{(m_i, t_m^i) \in \text{Data}|t_m^i \leq t \leq t_m^i + m_i/45\}$$

### A3.6 Interpolating Heart Rate and Step Count

We interpolate heart rate and step count using rolling window average with a window size of 5 minutes:

$$f_h(t) = \frac{1}{|H(t)|} \sum_{(h,t_h) \in H(t)} h, \quad H = \{(h_i, t_h^i) \in \text{Data}|t \leq t_h^i \leq t + 5\text{min}\}$$

$$f_v(t) = \frac{1}{|V(t)|} \sum_{(v,t_v) \in V(t)} v, \quad V = \{(v_i, t_v^i) \in \text{Data}|t \leq t_v^i \leq t + 5\text{min}\}$$

### A3.7 Choice of Time Grid and Discretization

Since our algorithm uses a forward-Euler style numerical integration scheme to solve the continuous neural ODE, we need to discretize the continuous, interpolated input features before we can use them. We first choose a suitable discrete time grid for each exercise instance. For each selected exercise instance, we focus on the time window from 210 minutes prior to the start of exercise, to 60 minutes after the start of exercise. We choose our discrete time grid as the CGM measurement time stamps

within the time window of interest, and since CGM measurements are consistently taken in 5 minute increments, this splits the time window into 5 minute intervals (with each interval containing one measurement) and we obtain 54 discrete time steps:

$$t_i = t_i^g = t_0^g + i\Delta t, \quad i = 0, \dots, 53, \quad \Delta t = 5\text{min}.$$

Given this time grid, the processed data used by all the models in the main paper are described in Table 3. Note that we do not interpolate CGM readings and use them as they are recorded, which is made possible by choosing the time grid to match the CGM recording time stamps $t_i = t_g^{i+48}$. In

| Feature | Definition |
|---|---|
| Time Stamp | $T_i = t_i = t_i^g$ |
| CGM Reading | $G_i = g_i$ |
| Average Insulin Injection Rate | $\overline{IIR}_i = \frac{1}{\Delta t} \int_{t_i}^{t_{i+1}} f_a(t) + f_b(t)\, dt$ |
| Average Carb Intake Rate | $M_i = \frac{1}{\Delta t} \int_{t_i}^{t_{i+1}} f_m(t)\, dt$ |
| Average Heart Rate | $H_i = f_h(t_i)$ |
| Average Step Count | $V_i = f_v(t_i)$ |

Table 3: Notations for various features

conclusion, each exercise instance, after above pre-processing, is turned into a 5-dimensional time series with 54 time steps of the form:

$$(G_i, \overline{IIR}_i, M_i, H_i, V_i)_{i=1}^{54}.$$

In Figure 4, we provide a graphical illustration of the data pre-processing pipeline.

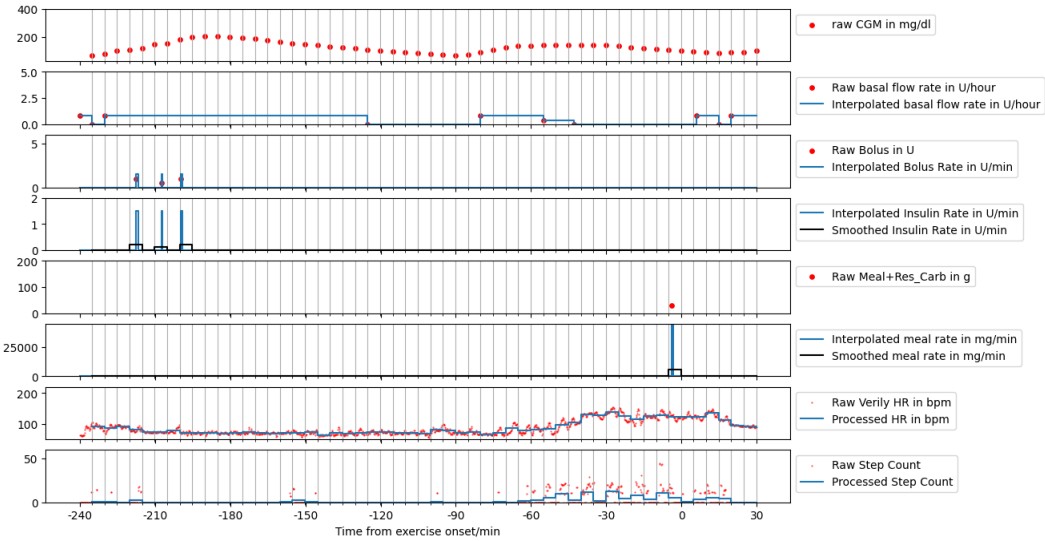

Figure 4: An illustration of raw and pre-process data of one of the exercise instances.

## A4 APPENDIX: EXPERIMENTAL DETAILS

### A4.1 SYNTHETIC MECHANISTIC GRAPH

In figure 5, we provide an illustration of the mechanistic graph used in the synthetic experiments.

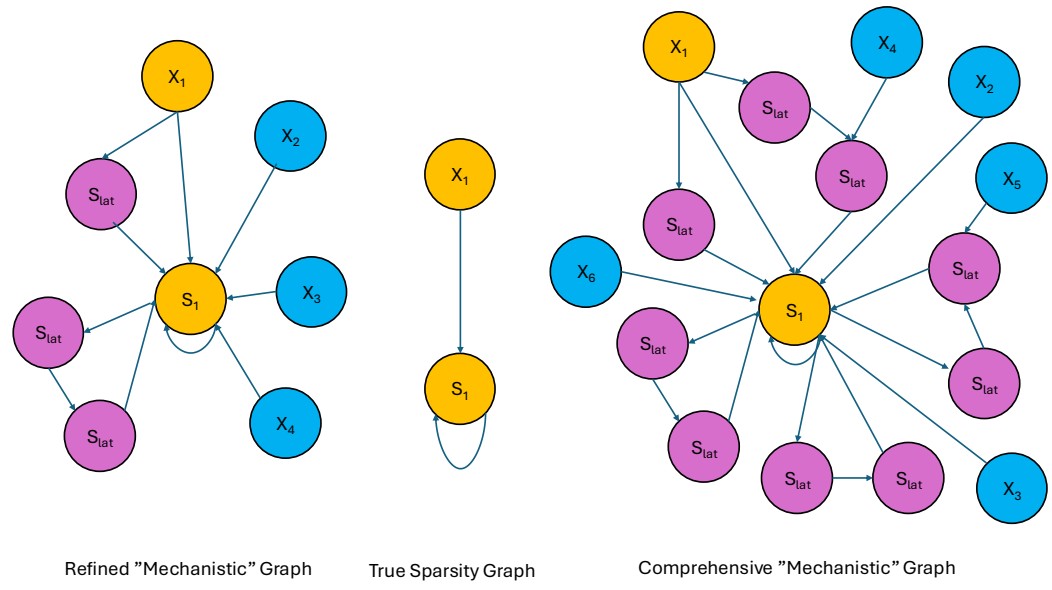

Figure 5: An illustration of the mechanistic vs true graphs used in the synthetic experiments

## A4.2 EXPERIMENTAL SET-UPS

**Synthetic data set-up** Synthetic data are generated with the following script:

```
def gen_syn_data(seed,exp=1,train_size=100):
rng=np.random.default_rng(seed=seed)
n=60
t=np.linspace(1,n,n).reshape(n,1)
data=[]
for k in range(train_size):
    x=[(i+1)/100*np.exp(1-t/n/10/(i+1))\
      +rng.normal(0,0.5,(n,1)) for i in range(1)]
    for i in range(3): // change to 6 for comprehensive graph
        x.append(rng.normal(0,0.5,(n,1)))
    x=np.concatenate(x,axis=1)
    dt=5e-2
    s1=[0]
    s2=[0]
    s3=[0]
    v=[0]
    for i in range(n):
        if exp==1:
            v.append(v[-1]+dt*(4*x[i,0]-0.5*(v[-1]-1)))
        else:
            v.append(v[-1]+dt*(4*x[i,0]-0.4*x[i,1]+0.04*x[i,2]\
                            -0.004*x[i,3]-0.5*(v[-1]-1)))
            // use v.append(v[-1]+dt*(4*x[i,0]-4e-1*x[i,1]\
                            +4e-2*x[i,2]-4e-3*x[i,3]\
                            +4e-4*x[i,4]-4e-5*x[i,5]\
```

```
                                          -0.5*(v[-1]-1)))
                                      for comprehensive graph
      sample=np.concatenate([np.reshape(v[1:],(n,1)),x],axis=-1)
      data.append(sample)
  cases=np.array(data)
  noise=rng.standard_normal(size=cases.shape,dtype='float64')
  cases=np.concatenate([cases,noise[:,:,:1]],axis=-1)
  return  cases
```

**Real-world data set-up**  For each model mentioned in the experiment section, here we offer a detailed description of the corresponding computational method and the hyper-parameters used. Throughout this section we are given an exercise time series of 54 time steps (corresponding to 54 5 minute intervals that made up the time window starting from 210 minutes prior to exercise termination, and ending at 30 minutes after exercise termination) and 5 features (corresponding to CGM reading, insulin, carbohydrate, heart rate, and step count, in that order). We denote the first feature (CGM reading) as $s_1$, and the other 4 features as $x$. We use superscript to indicate discrete time steps and subscript to indicate feature indices. For example, $x_1^{t_2}$ is the 1st input feature of $x$ (carbohydrate) at discrete time step $t = t_2$.

Our goal is to predict the CGM trace during the first 60 minutes following exercise onset corresponding to the output $s_1^{1:12} \in \mathbb{R}^1 2$ and therefore we set the number of prediction steps $q$ to be 12 for all models. We further split the given time series into historical context $D^P = (S_{\text{obs}}^P, X^P)^{t-41:t-1} \in \mathbb{R}^{41 \times 5}$, starting glucose $s_1^{t_0} \in \mathbb{R}$, inputs during exercise $X^{t_{0:11}} \in \mathbb{R}^{12 \times 4}$ (twelve inputs that are recorded 1 time step ahead of the expected outputs). We use $\hat{s}_1^F \in \mathbb{R}^{12}$ to indicate the CGM trace predicted by models, $s$ for model states and $z$ for black-box latent states, $h, c$ for the final hidden state and cell state of the LSTM initial condition learner. For ease of computation and without loss of generality, we set the $\Delta t$ term in forward-Euler style discretization to be 1 for all relevant models, and thus we omit it in the equations.

**Data Splits**  Since synthetic data are generated independently, each training set is split simply by the default index into 4 subsets for cross-validation, and the first split is used for re-training and obtaining the final model after the optimal hyper-parameter setting has been obtained. For real-world data, since there might be correlation between adjacent samples in the original pre-processed data, for each repetition, we generated a random permutations with:

```
rng=np.random.default_rng(seed=2024)
perms=np.zeros((repeats,cases.shape[0]),dtype='int32')
for i in range(repeats):
    perms[i]=rng.permutation(cases.shape[0])
print(cases.shape)
```

and apply the permutation before the standard index-based 3-fold train-validation split. Again, the first split is used for obtaining the final model with optimal hyper-parameter setting.

**Why Instance Level Random Permutation**  In real-world deployment, the model is expected to be applied to both existing patients (with new instances) and new patients, making cross-validation based on random instance-level splitting more appropriate, as it naturally includes predictions for both seen and unseen patients. In contrast, splitting strictly at the patient level would ignore the model's ability to generalize within the same patient across different instances—a nontrivial and practically important challenge in diabetes modeling. Furthermore, it is well recognized that *intra-patient* variability—the variation in glucose dynamics across different instances of the same

patient—can be as high as *inter-patient* variability in the diabetes research community. In other words, predicting future glucose trajectories for an existing patient can be as challenging as predicting the glucose level for a new, unseen patient. To quantify this phenomenon, we computed the root-mean-squared difference (RMSD) of mean and standard deviation of glucose levels both within and across patients in our data. Let $e_i$ and $s_i$ denote the mean and standard deviation of glucose in instance $i$, and define $N_{\text{inter}} = \{(i,j) \mid i < j,\ \text{instances } i \text{ and } j \text{ are from different patients}\}$ and $N_{\text{intra}} = \{(i,j) \mid i < j,\ \text{instances } i \text{ and } j \text{ are from the same patient}\}$. The intra-patient RMSD of mean glucose values, i.e., $\sqrt{\text{ave}_{(i,j)\in N_{\text{intra}}}(e_i - e_j)^2}$, was $54.24$, compared to an inter-patient RMSD of $55.10$, i.e., $\sqrt{\text{ave}_{(i,j)\in N_{\text{inter}}}(e_i - e_j)^2}$. Similarly, the intra- and inter-patient RMSD of glucose standard deviations ($\sqrt{\text{ave}_{(i,j)\in N_{\text{intra}}}(s_i - s_j)^2}$ and $\sqrt{\text{ave}_{(i,j)\in N_{\text{inter}}}(s_i - s_j)^2}$) were $22.75$ and $22.67$, respectively. Those summaries empirically confirming that intra-patient variability is comparable to inter-patient variability and result from patient-level cross-validation would likely be similar to that from instance-level cross-validation.

**Initialization and Optimizer** For all experiments, we use the Adam optimizer (Kingma & Ba, 2015) to perform stochastic gradient descent. We initialize all model weights with the PyTorch default setting with seed $2024 + r$, where $r$ increases by 1 starting from 0 for each experiment repetition (40 repeats for synthetic and 10 repeats for real-world experiments) unless state otherwise in the detailed model computation algorithm below.

**Training Epochs** In synthetic experiments, we train for 600 epochs and pick the epoch with best validation loss. For real-world experiment, We train for 200 epochs and pick the epoch with best validation loss.

**Hyper-parameter Search** We use grid search to tune hyper-parameters including learning rate and dropout rate. When choosing the grid, we restrict the search space to areas where the models have less than 20,000 parameters and we also try to limit the number of grid points to be less than 50. This is to make sure the computational cost of the experiments is capped at a reasonable level for small data sets and individual users. The grid used for each model in each experiment will be provided below together with model descriptions.

**Weight-sharing** We also enforced a constraint to simplify the optimization: if a latent node $v$ has only one incoming edge and one out-going edge, i.e., $u_1 \rightarrow v \rightarrow u_2$, then those two edges share a common weight $w_{u_1,v} = w_{v,u_2}$.

**Computing Resources and Time Usage** All experiments are performed on machines with Ubuntu 22.04 operating system, Xeon Gold 6148 CPU and single Nvidia 2080 Ti RTX 11GB GPU. Wall-clock run-time for both real-world and synthetic experiments range from 2 hour per repetition to 12 hours per repetition, depending on the size of the starting graph and the type of algorithm used, with MNODE_GL and S4D being the fastest ones and MNODE_GD and transformers being the slowest ones.

### A4.3 MNODE without Reduction (MNODE_NR)

We take the directed graph representation of $M_{\text{uva}}$, denoted as $G_{\text{uva}}$ to construct MNODE_NR.

**Hyper-parameters for real-world Experiments** The LSTM has 2 layers and $|V_{\text{uva}}|$ hidden dimension (i.e. this corresponds to the number of nodes in the directed graph) and for the same reason we set the 1st features of the initial condition state vector to be observed initial glucose level $s_1^{t_0}$. All MLPs have 2 hidden layers and 16 hidden units with dropout 0 and activation ReLu. We place $L_2$

---

**Algorithm 2** MNODE_NR

---

**Input:** historical context $D^{\mathrm{P}}$, starting glucose $s_1^{t_0}$, exogenous inputs $X^{t_{0:q-1}}$, the directed graph representation (as defined in Section 2) of the UVA-Padova model $G_{\mathrm{uva}}$, $\Delta t = 1$
$h, c = \mathrm{LSTM}(D^{\mathrm{P}})$
$\hat{S}^{t_0} = h[0]$
$\hat{S}_1^{t_0} = s_1^{t_0}$
**for** $i = 0 : q - 1$ **do**
$\quad s_{i+1} = s_i + \Delta t \cdot$
$\quad \hat{y}_{i+1} = s_{t+1}^1$
**end for**
**for** $i = 1 : q$ **do**
$\quad \hat{S}^{t_i} = \hat{S}^{t_{i-1}} + \Delta t \cdot \mathrm{MLPs}(S_i^{t_{i-1}}, X_i^{t_{i-1}}; G_{\mathrm{uva}})$
**end for**
**Output:** $\hat{s}_1^{t_{1:q}}$

---

regularization on all MLP parameters with penalty hyper-parameter $\lambda_2$, and train the model with learning rate $lr$. We tune these hyper-pameters with grid search on the following grid:

$$\lambda_2 = \{10^{-i} \mid i = 3, \ldots, 8\} \times lr = \{10^{-2}, 10^{-3}\}.$$

### A4.4   MNODE REDUCED BY DOMAIN KNOWLEDGE (MNODE_DK)

We take the reduced UVA-Padova model from Appendix C of Zou et al. (2024), and use its directed graph representation denoted as $G_{\mathrm{ruva}}$ to construct MNODE_DKR. Other than the graph used by MNODE, model computation is exactly the same the MNODE_NR.

**Hyper-parameters for real-world Experiments**   The LSTM has 2 layers and $|V_{\mathrm{ruva}}|$ hidden dimension (i.e. this corresponds to the number of nodes in the reduced graph) and for the same reason we set the 1st features of the initial condition state vector to be observed initial glucose level $s_1^{t_0}$. All MLPs have 2 hidden layers and 16 hidden units with dropout 0 and activation ReLu. We place $L_2$ regularization on all MLP parameters with penalty hyper-parameter $\lambda_2$, and train the model with learning rate $lr$. We tune these hyper-pameters with grid search on the following grid:

$$\lambda_2 = \{10^{-i} \mid i = 3, \ldots, 8\} \times lr = \{10^{-2}, 10^{-3}\}.$$

### A4.5   MNODE WITH HYBRID GRAPH SPARSIFICATION (MNODE_HGS)

We omit the implementation detail of MNODE_HGS here as it is already discussed in great detail in section 2.

**Hyper-parameters for real-world Experiments**   The LSTM has 2 layers and $|V_{\mathrm{uva}}^a|$ hidden dimension (i.e. this corresponds to the number of nodes in $G_{\mathrm{uva}}^{a,c}$) and for the same reason we set the 1st features of the initial condition state vector to be observed initial glucose level $s_1^{t_0}$. All MLPs have 2 hidden layers and 16 hidden units with dropout 0 and activation ReLu. We place $L_1$ regularization on all the edge weights with penalty hyper-parameter $\lambda_1$, $L_2$ regularization on all MLP parameters with penalty hyper-parameter $\lambda_2$, and train the model with learning rate $lr$. We tune these hyper-pameters with grid search on the following grid:

$$\lambda_1 = \{10^{-5}, 10^{-6}, 10^{-7}\} \times \lambda_2 = \{10^{-i} \mid i = 6, \ldots, 8\} \times lr = \{10^{-2}, 10^{-3}\}.$$

**Synthetic Experiments**   We also set $\hat{s}_1^{t_0} = s_1^{t_0}$ and uses the modified graph obtained from the given mechanistic graph. We keep everything else the same and the hyper-parameter search grid is:

$$\lambda_1 = \{10^{-6}, 10^{-7}\} \times \lambda_2 = \{10^{-i} \mid i = 6, \ldots, 8\} \times lr = \{10^{-2}, 10^{-3}\}.$$

## A4.6   MNODE reduced by exclusive group LASSO (MNODE_EGL)

MNODE_EGL uses the same model architecture as MNODE_HGS except (1) MNODE_EGL uses $G_{\text{uva}} = (V_{\text{uva}}, E_{\text{uva}})$ instead of $G_{\text{uva}}^{a,c}$, (2) the regularization term is defined as:

$$\lambda \sum_{v \in V_{\text{uva}}} \left( \sum_{(u,v) \in E_{\text{uva}}} |w_{(u,v)}| \right)^2$$

We tune hyper-pameters with grid search on the following grid:

$$\lambda = \{10^{-i} \mid i = 3, \ldots, 8\} \times lr = \{10^{-2}, 10^{-3}\}.$$

## A4.7   MNODE reduced by elastic net (MNODE_EN)

MNODE_EN uses the same model architecture as MNODE_HGS except (1) MNODE_EGL uses $G_{\text{uva}} = (V_{\text{uva}}, E_{\text{uva}})$ instead of $G_{\text{uva}}^{a,c}$, (2) the regularization term is defined as:

$$\sum_{(u,v) \in E_{\text{uva}}} \lambda_1 |w_{(u,v)}| + \lambda_2 w_{(u,v)}^2$$

**Hyper-parameters for real-world Experiments**   We tune these hyper-pameters with grid search on the following grid:

$$\lambda_1 = \{10^{-5}, 10^{-6}, 10^{-7}\} \times \lambda_2 = \{10^{-i} \mid i = 6, \ldots, 8\} \times lr = \{10^{-2}, 10^{-3}\}.$$

## A4.8   MNODE reduced by Neural Sparse (MNODE_NS)

The neural sparse algorithm tries to learn a distribution (parameterized by a neural networks) form which good sub-graphs are samples. Its computation is given below:

---
**Algorithm 3** MNODE_NS

---
**Input:** historical context $D^{\text{P}}$, starting glucose $s_1^{t_0}$, exogenous inputs $X^{t_{0:q-1}}$, the directed graph representation (as defined in Section 2) of the UVA-Padova model $G_{\text{uva}}$, $\Delta t = 1$, $K$ size of the sub-graphs to be sampled
Index edges in $E_{\text{uva}}$ by some order
Initialize model parameter $\alpha \sim \mathcal{N}(0, 1), \quad \alpha \in \mathbb{R}^{|E_{\text{uva}}|}$
Initialize model parameter $\epsilon \sim \mathcal{N}(0, 1), \quad \epsilon \in \mathbb{R}^{K \times |E_{\text{uva}}|}$
$\pi = \text{softmax}(\alpha)$
$w = \exp\left((\log(\pi) - \log(-\log(\epsilon))) \cdot 10\right)$
$w = \frac{w}{\sum_i w_i}$
Round $w$ to 2 decimal places
Construct $E'_{\text{uva}} = \{e_i \in E_{\text{uva}} \mid w_i > 0\}$, and construct $G'_{\text{uva}} = (V_{\text{uva}}, E'_{\text{uva}})$
Run MNODE_NR with $D^{\text{P}}$, $s_1^{t_0}$, $X^{t_{0:q-1}}$, $G'_{\text{uva}}$ and return its output

---

**Hyper-parameters for real-world Experiments**   The hyper-parameters and model configuration for the MNODE_NR part of the model is identical to the implementation of MNODE_NR above. In addition to $\lambda_2$ and $lr$, we tune $K$, the maximal number of edges to be included in the sampled sub-graph. These hyper-parameters are tuned with grid search on the following grid:

$$\lambda_2 = \{10^{-i} \mid i = 3, \ldots, 8\} \times lr = \{10^{-2}, 10^{-3}\} \times K = \{12, 16, 20, 24\}.$$

**Hyper-parameters for Synthetic Experiments**   We keep everything else the same and the hyper-parameter search grid is:

$$\lambda_2 = \{10^{-3}\} \times lr = \{10^{-2}, 10^{-3}\} \times K = \{2, 4, \ldots, 10\}.$$

### A4.9  MNODE REDUCED BY GREEDY SEARCH (MNODE_GD)

The greedy search is implemented in the standard step-wise backward way: at each iteration, the algorithm considers all existing edges and choose the one whose removal leads to the most improvement in validation loss, and stop when no edge's removal improves validation loss. Specifically: Since

---

**Algorithm 4** MNODE_GD

---

**Input:** historical context $D^{\mathrm{P}}$, starting glucose $s_1^{t_0}$, exogenous inputs $X^{t_0:q-1}$, the directed graph representation of the UVA-Padova model $G_{\mathrm{uva}}$, $\Delta t = 1$
Index edges $e$ in $E_{\mathrm{uva}}$ by some order
Stop=FALSE
MinLoss$= 10^7$
**while** Stop = FALSE **do**
  **for** $e_i \in E_{\mathrm{uva}}$ **do**
    Construct $G' = (V_{\mathrm{uva}}, E_{\mathrm{uva}} \setminus \{e_i\})$
    Run MNODE_NR with $D^{\mathrm{P}}$, $s_1^{t_0}$, $X^{t_0:q-1}$, $G'$, record validation loss as $l_i$
  **end for**
  **if** $\min(\{l_i\}_{i=1}^{|E_{\mathrm{uva}}|}) <$ MinLoss **then**
    MinLoss$= \min(\{l_i\}_{i=1}^{|E_{\mathrm{uva}}|})$
    $E_{\mathrm{uva}} = E_{\mathrm{uva}} \setminus \{e_{\mathrm{argmin}\{l_i\}}\}$
  **else**
    Stop=TRUE
  **end if**
**end while**
Construct $G^* = (V_{\mathrm{uva}}, E_{\mathrm{uva}})$
Run MNODE_NR with $D^{\mathrm{P}}$, $s_1^{t_0}$, $X^{t_0:q-1}$, $G^*$ and return its output

---

the greedy algorithm can be extremely slow, we do not tune hyper-parameters and instead fix the $L_2$ penalty hyper-parameter to be $10^{-6}$ and learning rate to be $10^{-3}$. All other settings are the same as MNODE_NR.

### A4.10  MNODE REDUCED BY RANDOM SEARCH (MNODE_RD)

The random search randomly picks 5 sub-graphs that has contains $1 - p$ percent of edges and select the best one. Specifically:

---

**Algorithm 5** MNODE_RD

---

**Input:** historical context $D^{\mathrm{P}}$, starting glucose $s_1^{t_0}$, exogenous inputs $X^{t_0:q-1}$, the directed graph representation of the UVA-Padova model $G_{\mathrm{uva}}$, number of random sub-graphs $R = 5$, sub-graph edge ratio $P = \{0.1, 0.2, 0.4\}$, $\Delta t = 1$
**for** $p \in P$ **do**
  Uniformly sample $R$ sub-graphs of $G_{\mathrm{uva}}$ that have the same number of nodes and $(1-p)|E_{\mathrm{uva}}|$ number of edges, denote them as $G_{p,1}, \ldots, G_{p,R}$
  **for** $r = 1 : R$ **do**
    Run MNODE_NR with $D^{\mathrm{P}}$, $s_1^{t_0}$, $X^{t_0:q-1}$, $G_{p,r}$, record validation loss as $l_{p,r}$
  **end for**
**end for**
Return the output of MNODE_NR with $G_{\mathrm{argmin}\{l_{p,r}\}}$

---

**Hyper-parameters for real-world Experiments**  As in greedy search, random search is also slow on real-world data. Therefore, we do not tune hyper-parameters and instead fix the $L_2$ penalty hyper-parameter to be $10^{-3}$ and learning rate to be $0.02$. All other settings are the same as MNODE_NR.

**Hyper-parameters for Synthetic Experiments** In the synthetic case, the graph is smaller and we can afford to tune $\lambda_2$ over $\{10^{-6}\}$. The learning rate is fixed to $10^{-3}$. All other settings are the same as real-data.

## A4.11 BNODE

For BNODE and the subsequent black-box models, we point the reader to implementations referenced in the associated citations in the main paper. Here we describe our implementation.

---

**Algorithm 6** Black-box Neural ODE Model

---

**Input:** historical context $D^{\mathrm{P}}$, starting glucose $s_1^{t_0}$, exogenous inputs $X^{t_{0:q-1}}$, $\Delta t = 1$
$h, c = \mathrm{LSTM}(D^{\mathrm{P}})$
$\hat{S}^{t_0} = h[0]$
$\hat{S}_1^{t_0} = s_1^{t_0}$
**for** $i = 1 : q$ **do**
$\quad \hat{S}^{t_i} = \hat{S}^{t_{i-1}} + \Delta t \cdot \mathrm{MLPs}(S^{t_{i-1}}, X^{t_{i-1}})$
**end for**
**Output:** $\hat{s}_1^{t_{1:q}}$

---

**Hyper-parameters for real-world Experiments** The LSTM has 2 layers and $d$ hidden dimension. Note that here the hidden dimension of LSTM also determines the state dimension of the neural ODE, which is a tunable hyperparameter. All MLPs have 2 hidden layers and 16 hidden units with dropout $a$ and activation ReLU, trained with learning rate of $lr$ we tune these hyper-parameters with grid search on the following grid:

$$d = \{6, 12, 18\} \times a = \{0, 0.1, 0.2\} \times lr = \{10^{-2}, 10^{-3}, 10^{-4}\}.$$

**Hyper-parameters for Synthetic Experiments** We keep all the settings the same as the real-world experiments

## A4.12 TCN

---

**Algorithm 7** Temporal Convolutional Network Model

---

**Input:** historical context $D^{\mathrm{P}}$, starting glucose $s_1^{t_0}$, exogenous inputs $X^{t_{0:q-1}}$,
$\tilde{X}^{t_{0:q=1}} = \mathbf{0} \in \mathbb{R}^q$
$\tilde{X}^{t_0} = s_1^{t_0}$
$X' = \mathrm{concatenate}(\tilde{X}, X, \dim = \text{features})$
$seq_{in} = \mathrm{concatenate}(D^{\mathrm{P}}, X', \dim = \text{time})$
$seq_{out} = \mathrm{TCN}(seq_i n)$
$\hat{s}_1^{t_{1:q}} = \mathrm{Linear}(seq_{out})$
**Output:** $\hat{s}_1^{t_{1:q}}$

---

**Hyper-parameters for real-world Experiments** The TCN model is taken directly from the code repository posted on `https://github.com/locuslab/TCN/blob/master/TCN/tcn.py`, with input size set to 5, number of channels set to a list of $n$ copies of $m$, kernel size set to $l$ and dropout set to $a$, trained with learning rate $lr$. We tune these hyper-parameters with grid search on the following grid:

$$n = \{2, 3\} \times m = \{16, 32\} \times l = \{2, 3, 4\} \times a = \{0, 0.1, 0.2\} \times lr = \{10^{-2}, 10^{-3}, 10^{-4}\}.$$

## A4.13 LSTM

---

**Algorithm 8** Long Short Term Memory Model

---

**Input:** historical context $D^{\text{P}}$, starting glucose $s_1^{t_0}$, exogenous inputs $X^{t_0:q-1}$,
$h, c = $ Encoder LSTM$(D^{\text{P}})$
Set initial hidden state and cell state of Decoder LSTM to $h, c$ respectively
$seq_{out}, h_q, c_q = $ Decoder LSTM$(X^{t_0:5})$
$\hat{s}_1^{t_1:q} = $ Linear$(seq_{out})$
**Output:** $\hat{s}_1^{t_1:q}$

---

**Hyper-parameters for real-world Experiments**    Both Encoder and Decoder LSTM have $n$ layers and $d$ hidden states with dropout set to $a$, trained with learning rate $lr$. We tune these hyper-parameters with grid search on the following grid:

$$n = \{2, 3\} \times m = \{6, 12, 18\} \times a = \{0, 0.1, 0.2\} \times lr = \{10^{-2}, 10^{-3}, 10^{-4}\}.$$

**Hyper-parameters for Synthetic Experiments**    The settings are the same as real-world experiments.

### A4.14   TRANSFORMER

---

**Algorithm 9** Transformer Model

---

**Input:** historical context $D^{\text{P}}$, starting glucose $s_1^{t_0}$, exogenous inputs $X^{t_0:q-1}$, true output $s_1^{t_1:q-1}$ (needed during training)
$\tilde{X} = \mathbf{0} \in \mathbb{R}^q$
$\tilde{X}^{t_0} = s_1^{t_0}$
$X' = $ concatenate$(\tilde{X}, X, \dim = \text{features})$
encoder_in $=$ concatenate$(D^{\text{P}}, X', \dim = \text{time})$ (concatenating all inputs to form a masked context)
**if** Model in Training Mode **then**
    decoder_in $=$ concatenate$(s_1^{t_0}, s_1^{t_1:q-1})$ (expected output shifted to the right)
    decode_out $=$ Transformer(encoder_in, decoder_in, decoder_causal_mask)
**end if**
**if** Model in Evaluation Mode **then**
    decoder_in $=$ concatenate$(s_1^{t_0}, \mathbf{0} \in \mathbb{R}^{q-1})$
    **for** $i = 1 : q - 1$ **do**
       decode_out $=$ Transformer(encoder_in, decoder_in)
       decoder_in$_{i+1} = $ decode_out$_i$
    **end for**
    decode_out $=$ Transformer(encoder_in, decoder_in)
**end if**
$\hat{s}_1^{t_1:q} = $ Linear(decode_out)
**Output:** $\hat{s}_1^{t_1:q}$

---

**Hyper-parameters for real-world Experiments**    We use the transformer model provided by the pytorch nn class, and its hyper-parameters are set as follows: d_model set to $d$, number of attention heads set to $n$, number of encoder layers set to 2, number of decoder layers set to 2, the dim_feedforward is set to $m$ and dropout is set to $a$, trained at a learning rate of $10^{-3}$. We tune the hyper-parameters with the following grid:

$$d = \{8, 16\} \times n = \{4, 8\} \times m = \{16, 32\} \times a = \{0, 0.1\}$$

## A4.15   S4D

---

**Algorithm 10** S4 Diagonal Model

---

**Input:** historical context $D^{\mathrm{P}}$, starting glucose $s_1^{t_0}$, exogenous inputs $X^{t_{0:q-1}}$
$\tilde{X} = \mathbf{0} \in \mathbb{R}^q$
$\tilde{X}^{t_0} = s_1^{t_0}$
$X' = \mathrm{concatenate}(\tilde{X}, X, \dim = \mathrm{features})$
$seq_{in} = \mathrm{concatenate}(D^{\mathrm{P}}, X', \dim = \mathrm{time})$
$seq_{in} = \mathrm{Linear}(Seq_{in})$
$seq_{in} = \mathrm{Transpose}(Seq_{in}, 1, 2)$
$seq_{out} = \mathrm{S4D}(seq_i n)$
$seq_{out} = \mathrm{Transpose}(Seq_{out}, 1, 2)$
$\hat{s}_1^{t_{1:q}} = \mathrm{Linear}(seq_{out})_{-q:}$
**Output:** $\hat{s}_1^{t_{1:q}}$

---

**T1DEXI Experiments**   We take the S4D model directly from the following github repository `https://github.com/thjashin/multires-conv/blob/main/layers/s4d.py`, and its hyper-parameters are set as: d_model set to $d$, d_state set to $m$, dropout set to $a$, trained at a learning rate of $lr$. We tune the hyper-parameters with the following grid:

$$d = \{4, 6, 8\} \times \{m = \{32, 64\} \times a = \{0, 0.1, 0.2\} \times lr = \{10^{-2}, 10^{-3}, 10^{-4}\}$$

## A5   APPENDIX: EVALUATION

In this section we describe how we evaluate the performance of various learning algorithms. Suppose our training data $D = \{z_1, \ldots, z_N\} \subset \mathcal{Z}$ are i.i.d. samples from some distribution $P$, and a learning algorithm $M : \mathcal{Z} \to \mathcal{F}$ maps $D$ of arbitrary size to a function $f_D$. We also have an evaluation loss function (which may not be the training loss function) $L : \mathcal{F} \times \mathcal{Z} \to \mathbb{R}^+ \cup \{0\}$ that maps a learned function and a test sample to a non-negative value. We are primarily interested in evaluating the expected prediction error (also known as generalization error) of the algorithms on unseen training and test data:

$$\mathrm{EPE}(M, L, P) = \mathbb{E}_{Z \sim P, D \sim P}[L(\hat{f}_D, Z)].$$

In our setting, $Z$ can be written as $(S_{\mathrm{obs}}^{a,\mathrm{P}}, X^{a,\mathrm{P}}, X^{a,\mathrm{F}}, S_{\mathrm{obs}}^{a,\mathrm{F}})$ and the learned function maps the input $(S_{\mathrm{obs}}^{a,\mathrm{P}}, X^{a,\mathrm{P}}, X^{a,\mathrm{F}})$ to the output $\hat{f}_D(S_{\mathrm{obs}}^{a,\mathrm{P}}, X^{a,\mathrm{P}}, X^{a,\mathrm{F}})$ to approximate $S_{\mathrm{obs}}^{a,\mathrm{P}}$.

### A5.1   SYNTHETIC EXPERIMENTS

In the synthetic experiment, we first generate a test set $\tilde{D} = \{\tilde{z}_1, \cdots, \tilde{z}_M\}$ with a sufficiently large sample size $M = 10000$, where $\tilde{z}_1, \cdots, \tilde{z}_M$ are i.i.d samples from distribution $P$. We then generate $K = 40$ copies of training data, $D_k, k = 1, \cdots K$, of size $N = 100/1000$ each, perform the experiment $K$ rounds with the $K$ training sets and $K$ different random seeds, and obtain the corresponding prediction function $\hat{f}_{D_k}$. We choose $L$ to be the standard mean squared error loss function and our RMSE estimator for the k-th experiment round is computed as:

$$\widehat{\mathrm{RMSE}}_k = \sqrt{\frac{1}{M} \sum_{m=1}^{M} \|\hat{f}_{D_k}(S_{\mathrm{obs},m}^{a,\mathrm{P}}, X_m^{a,\mathrm{P}}, X_m^{a,\mathrm{F}}) - S_{\mathrm{obs},m}^{a,\mathrm{F}}\|_2^2}.$$

The reported RMSE is the average RMSE over the $K$ rounds:

$$\widehat{\mathrm{RMSE}} = \frac{1}{K} \sum_{k=1}^{K} \widehat{\mathrm{RMSE}}_k,$$

and the $1\sigma$ Monte carlo standard error is computed as

$$\widehat{\text{s.e.}} = \sqrt{\frac{1}{K(K-1)} \sum_{k=1}^{K} (\widehat{\text{RMSE}} - \widehat{\text{RMSE}}_k)^2}.$$

## A5.2 REAL WORLD EXPERIMENT

In this section we describe how we evaluate the performance of a learning algorithm using observed data only.

Note that when the loss is the mean squared error, we have the following variance-bais decomposition of EPE:

$$\text{EPE} = \mathbb{E}_{Z,D}[\|f_D(S_{\text{obs}}^{a,\text{P}}, X^{a,\text{P}}, X^{a,\text{P}}) - S_{\text{obs}}^{a,\text{P}}\|_2^2] = \text{variance} + \text{bias}^2 + \text{noise}$$

$$\text{variance} = \mathbb{E}_{Z,D} \left\{ \left\| \widehat{f}_D(S_{\text{obs}}^{a,\text{P}}, X^{a,\text{P}}, X^{a,\text{P}}) - \mathbb{E}_D[\widehat{f}_D(S_{\text{obs}}^{a,\text{P}}, X^{a,\text{P}}, X^{a,\text{P}})] \right\|_2^2 \right\}$$

$$\text{bias}^2 = \mathbb{E}_Z \left\{ \left\| \mathbb{E}_D[\widehat{f}_D(S_{\text{obs}}^{a,\text{P}}, X^{a,\text{P}}, X^{a,\text{P}})] - \mathbb{E}[S_{\text{obs}}^{a,\text{P}} \mid S_{\text{obs}}^{a,\text{P}}, X^{a,\text{P}}, X^{a,\text{P}}] \right\|_2^2 \right\}$$

$$\text{noise} = \mathbb{E}_Z \left\{ \left\| \mathbb{E}[S_{\text{obs}}^{a,\text{P}} \mid S_{\text{obs}}^{a,\text{P}}, X^{a,\text{P}}, X^{a,\text{P}}] - S_{\text{obs}}^{a,\text{P}} \right\|_2^2 \right\}.$$

In a synthetic experiment, one may sample from $P$ as much as possible and estimate all the above quantities as described in section A5.1. In reality, however, $P$ is unknown and we only have limited data. Therefore, we have to resort to techniques such as cross validation or bootstrap. While the standard $K$-fold cross validation can give unbiased estimator for EPE, it cannot estimate variance or bias because each sample only enters the test set once. Due to the implausibility of observing repeated samples with the same input in this case, we make the ideal assumption that there is no noise in predicting state variables, i.e.,

$$\mathbb{E}[S_{\text{obs}}^{a,\text{P}}|S_{\text{obs}}^{a,\text{P}}, X^{a,\text{P}}, X^{a,\text{P}}] = S_{\text{obs}}^{a,\text{P}},$$

so that we can estimate bias and variance separately. Under this assumption, noise is zero and $\text{EPE} = \text{bias}^2 + \text{variance}$. Specifically, in this paper we use a modified K-fold cross-validation to construct unbiased estimators for variance and bias. First, split $D$ into $K$ disjoint subsets of equal size $D = \bigsqcup_{k=1}^{K} D_k$, denote $D^{(-k)} = D \setminus D_k$. Then, each $D^{(-k)}$ consists of $K-1$ disjoint subsets $D_i$ :

$$D^{(-k)} = \bigsqcup_{i \in I_K^{(-k)}} D_i,$$

where $I_K^{(-k)} = \{i \mid 1 \le i \le K, i \ne k\}$. Our estimator for variance is:

$$\widehat{\text{variance}} = \frac{1}{|D|} \sum_{k=1}^{K} \sum_{(S,X) \in D_k} \frac{1}{K-1} \sum_{i \in I_K^{(-k)}} \left\| \widehat{f}_{D_i}(S_{\text{obs}}^{a,\text{P}}, X^{a,\text{P}}, X^{a,\text{P}}) - \overline{f_{D^{(-k)}}(S_{\text{obs}}^{a,\text{P}}, X^{a,\text{P}}, X^{a,\text{P}})} \right\|_2^2$$

$$\overline{f_{D^{(-k)}}(S_{\text{obs}}^{a,\text{P}}, X^{a,\text{P}}, X^{a,\text{P}})} = \frac{1}{K-1} \sum_{i \in I_K^{(-k)}} \widehat{f}_{D_i}(S_{\text{obs}}^{a,\text{P}}, X^{a,\text{P}}, X^{a,\text{P}})$$

$$\widehat{\text{bias}}^2 = \frac{1}{|D|} \sum_{k=1}^{K} \sum_{(S,X) \in D_k} \left\| \overline{f_{D^{(-k)}}(S_{\text{obs}}^{a,\text{P}}, X^{a,\text{P}}, X^{a,\text{P}})} - S_{\text{obs}}^{a,\text{P}} \right\|_2^2$$

$$\widehat{\text{MSE}} = \widehat{\text{variance}} + \widehat{\text{bias}}^2 = \frac{1}{N(K-1)} \sum_{k=1}^{K} \sum_{(S,X) \in D_k} \sum_{i \in I_K^{(-k)}} \left\| \widehat{f}_{D_i}(S_{\text{obs}}^{a,\text{P}}, X^{a,\text{P}}, X^{a,\text{P}}) - S_{\text{obs}}^{a,\text{P}} \right\|_2^2.$$

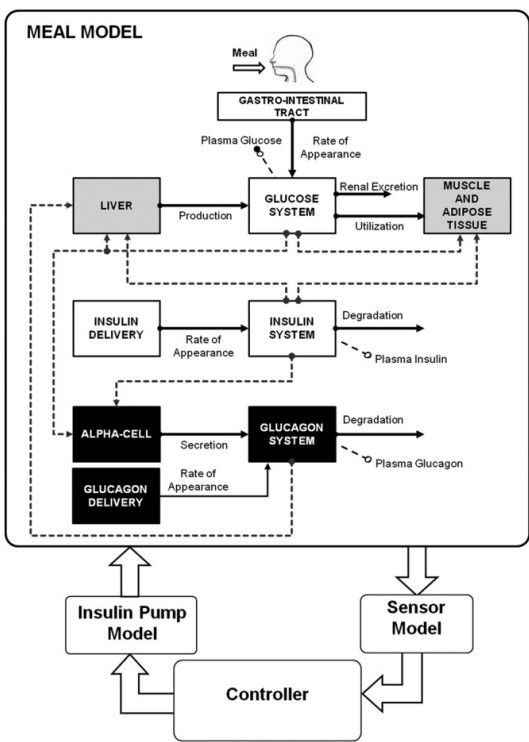

Figure 6: UVA/Padova Simulator S2013, taken from Figure 1 of Man et al. (2014)

$$\widehat{\text{RMSE}} = \sqrt{\widehat{\text{MSE}}}$$

We perform the experiment $R = 10$ rounds using the above procedure, each round with a different permutation of training data and a new random seed, to obtain $R$ estimates of variance and RMSE, and report their average. Note that this method effectively estimates the variance and bias of prediction algorithm trained with $N/K$ rather than $N$ samples. As a consequence, it tends to underestimate the prediction performance. Standard errors are computed in the same way by dividing standard deviation by the square root of $R$.

## A6   UVA-PADOVA SIMULATOR S2013

Here we provide the exact full UVA-Padova S2013 model equations. Variables that are not given meaningful interpretations are model parameters.

### A6.1   SUMMARY DIAGRAM

At a high level, UVA-Padova can be summarized by the diagram in Figure 6, which is taken from Figure 1 in Man et al. (2014). It divides the complex physiological system into 10 subsystems, which are linked by key causal states such as Rate of Appearance, Endogenous Glucose Production and Utilization. Next, we will introduce each subsystem one by one and also explain the physiological meanings behind state variables.

## A6.2  GLUCOSE SUBSYSTEM

$$\dot{G}_p = EGP + Ra - U_{ii} - E - k_1 G_p + k_2 G_t \tag{5}$$

$$\dot{G}_t = -U_{id} + k_1 G_p - k_2 G_t \tag{6}$$

$$G = G_p / V_G \tag{7}$$

$G_p$: Plasma Glucose, $G_t$ Tissue Glucose, $EGP$: Endogenous Glucose Production Rate, $Ra$ Rate of Glucose Appearance, $U_{ii}$: Insulin-independent Utilization Rate, $U_{id}$: Insulin-dependent Utilization Rate, $E$ Excretion Rate, $V_G$ Volume Parameter, $G$ Plasma Glucose Concentration

## A6.3  INSULIN SUBSYSTEM

$$\dot{I}_p = -(m_2 + m_4)I_p + m_1 I_l + Rai \tag{8}$$

$$\dot{I}_t = -(m_1 + m_3)I_t + m_2 I_p \tag{9}$$

$$I = I_p / V_I \tag{10}$$

$I_p$ Plasma Insulin, $I_l$ Liver Insulin, $Rai$ Rate of Insulin Appearance, $V_l$ Volume Parameter, $I$ Plasma Insulin Concentration

## A6.4  GLUCOSE RATE OF APPEARANCE

$$Q_{sto} = Q_{sto1} + Q_{sto2} \tag{11}$$

$$\dot{Q}_{sto1} = -k_{gri}Q_{sto1} + D \cdot \delta \tag{12}$$

$$\dot{Q}_{sto2} = -k_{empt}(Q_{sto}) \cdot Q_{sto2} + k_{gri}Q_{sto1} \tag{13}$$

$$\dot{Q}_{gut} = -k_{abs}Q_{gut} + k_{empt}(Q_{sto}) \cdot Q_{sto2} \tag{14}$$

$$Ra = f k_{abs}Q_{gut}/(BW) \tag{15}$$

$$k_{empt}(Q_{sto}) = k_{min} + (k_{max} - k_{min})(\tanh(\alpha Q_{sto} - \alpha bD) - \tanh(\beta Q_{sto} - \beta cD) + 2)/2 \tag{16}$$

$Q_{sto1}$: First Stomach Compartment, $Q_{sto2}$: Second Stomach Compartment, $Q_{gut}$: Gut Compartment, $\delta$ Carbohydrate Ingestion Rate

## A6.5  ENDOGENOUS GLUCOSE PRODUCTION

$$EGP = k_{p1} - k_{p2}G_p - k_{p3}X^L + \xi X^H \tag{17}$$

$$\dot{X}^L = -k_i(X^L - I_r] \tag{18}$$

$$\dot{I}_r = -k_i(I_r - I) \tag{19}$$

$$\dot{X}^H = -k_H X^H + k_H \max(H - H_b) \tag{20}$$

$X^L$: Remote Insulin Action on EGP, $X^H$: Glucagon Action on EGP, $I_r$ Remote Insulin Concentration, $H$ Plasma Glucagon Concentration, $H_b$: Basal Glucagon Concentration Parameter

## A6.6 GLUCOSE UTILIZATION

$$U_{ii} = F_{cns} \tag{21}$$

$$U_{id} = \frac{(V_{m0} + V_{mx}X(1 + r_1 \cdot risk))G_t}{K_{m0} + G_t} K_{m0} + G_t \tag{22}$$

$$\dot{X} = -p_{2U}X + p_{2U}(I - I_b) \tag{23}$$

$$risk = \begin{cases} 0 & G_b \leq G \\ 10(\log(G) - \log(G_b))^{2r_2} & G_{th} \leq G < G_b \\ 10(\log(G_{th}) - \log(G_b))^{2r_2} & G < G_{th} \end{cases} \tag{24}$$

$F_{cns}$: Glucose Independent Utilization Constant, $X$: Insulin Action on Glucose Utilization, $I_b$ Basal Insulin Concentration Constant, $risk$ Hypoglycemia Risk Factor, $G_b$ Basal Glucose Concentration Parameter, $G_{th}$ Hypoglycemia Glucose Concentration Threshold.

## A6.7 RENAL EXCRETION

$$\dot{E} = k_{e1} \max(G_p - k_{e2}, 0) \tag{25}$$

## A6.8 SUBCUTANEOUS INSULIN KINETICS

$$Rai = k_{a1}I_{sc1} + k_{a2}I_{sc2} \tag{26}$$

$$\dot{I}_{sc1} = -(k_d + k_{a1})I_{sc1} + IIR \tag{27}$$

$$\dot{I}_{sc2} = k_d I_{sc1} - k_{a2}I_{sc2} \tag{28}$$

$I_{sc1}$: First Subcutaneous Insulin Compartment, $I_{sc2}$: Second Subcutaneous Insulin Compartment, $IIR$ Exogenous Insulin Delivery Rate

## A6.9 SUBCUTANEOUS GLUCOSE KINETICS

$$\dot{G}_s = -T_s G_s + T_s G \tag{29}$$

$G_s$: Subcutaneous Glucose Concentration

## A6.10 GLUCAGON SECRETION AND KINETICS

$$\dot{H} = -nH + SR_H + Ra_H \tag{30}$$

$$SR_H = SR_H^s + SR_H^d \tag{31}$$

$$\dot{SR}_H^s = \begin{cases} -\rho \left[ SR_H^s - \max \left( \sigma_2(G_{th} - G) + SR_H^b, 0 \right) \right] & G \geq G_b \\ -\rho \left[ SR_H^s - \max \left( \frac{\sigma(G_{th} - G)}{I + 1} + SR_H^b, 0 \right) \right] & G < G_b \end{cases} \tag{32}$$

$$\dot{SR}_H^d = \eta \max(-\dot{G}, 0) \tag{33}$$

$SR_H^s$: First Glucagon Secretion Compartment, $SR_H^d$: Second Glucagon Secretion Compartment, $SR_H^b$: Basal Glucagon Secretion Parameter, $Ra_H$: Rate of Glucagon Appearance

## A6.11 SUBCUTANEOUS GLUCAGON KINETICS

$$\dot{H}_{sc1} = -(k_{h1} + k_{h2})H_{sc1} + H_{inf} \tag{34}$$

$$\dot{H}_{sc2} = k_{h1}H_{sc1} - k_{h3}H_{sc2} \tag{35}$$

$$Ra_H = k_{h3}H_{sc2} \tag{36}$$

$H_{sc1}$: First Subcutaneous Glucagon Compartment, $H_{sc2}$: Second Subcutaneous Glucagon Compartment, $H_{inf}$ Subcutaneous Glucagon Infusion Rate.

## A7 EXTRA RESULTS

### A7.1 ADDITIONAL METRICS FOR SYNTHETIC EXPERIMENTS

**Comments** As shown in Figure 7 to Figure 12, we observed similar trends and patterns in model performance for metrics including MAPE, Peak MAPE and Pearson Correlation as those observed in main text for RMSE.

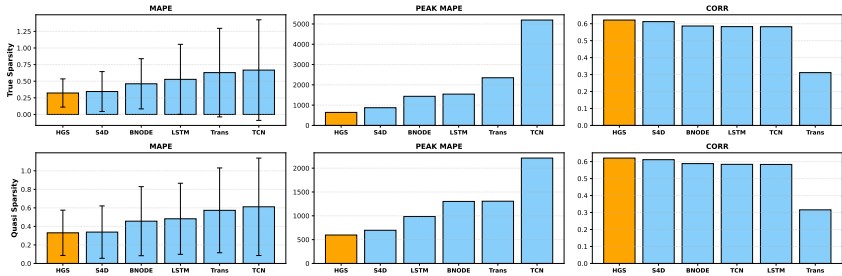

Figure 7: Comparing against black-box models, training size = 100

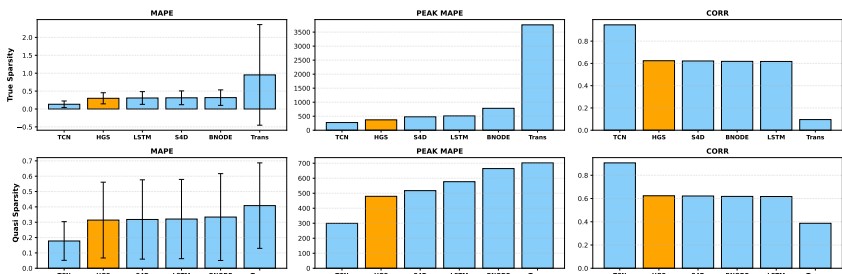

Figure 8: Comparing against black-box models, training size = 1000

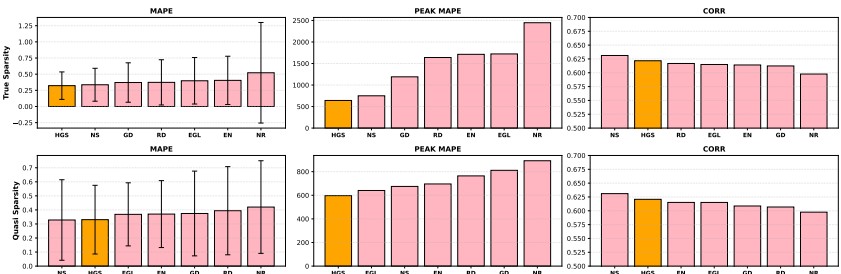

Figure 9: Comparing against other reduction methods, refined graph, training size = 100

### A7.2 ABLATION STUDY ON HGS UNDER LIMITED DATA, TRUE SPARSITY REGIME WITH COMPREHENSIVE STARTING GRAPH

**Comments** To better understand why regularization based methods perform poorly on comprehensive graph, we also performed ablation study on HGS model components and the results are shown in Figure 13. We can see that graph modification (step 1 + step 2) alone or regularization (step 3) alone

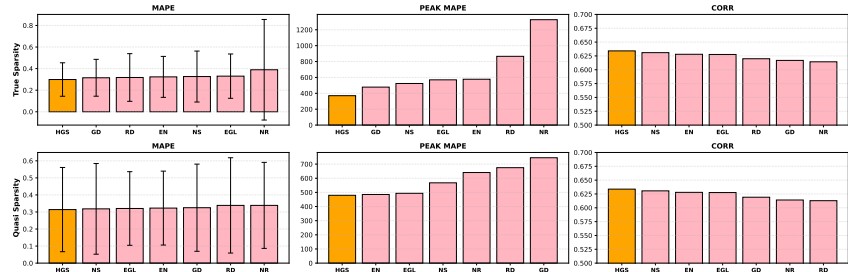

Figure 10: Comparing against other reduction methods, refined graph, training size = 1000

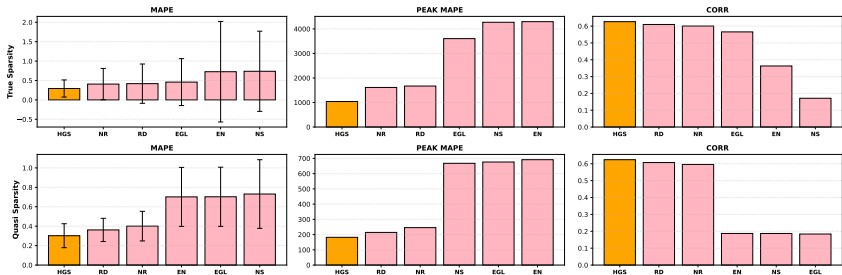

Figure 11: Comparing against other reduction methods, comprehensive graph, training size = 100

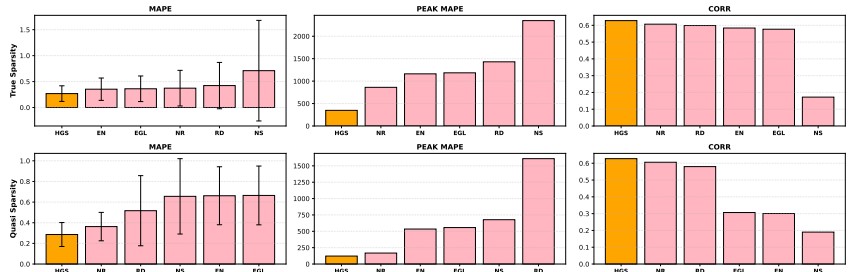

Figure 12: Comparing against other reduction methods, comprehensive graph, training size = 1000

is not effective (and even worse than no reduction) at improving predictive performance or robustness. It requires both to achieve the desired outcome.

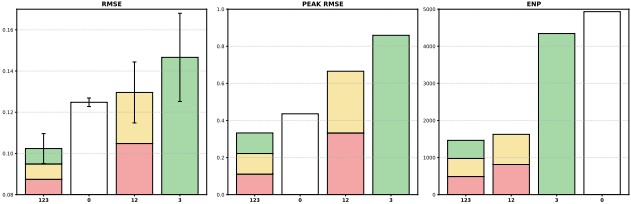

Figure 13: Ablation study on model components, training size = 100, true sparsity, comprehensive graph

## A8 INSTABILITY OF DYNAMICAL SYSTEMS WITH CYCLES: A TOY EXAMPLE

Here we discuss how cycles/loops in dynamical systems can lead to instability. There are three sources of numerical stability: blowing-up, exploding gradient and stiffness.

## A8.1 Blowing-up

Consider a simple 2 state system with a circular dependence:

$$\frac{ds_1(t)}{dt} = as_1(t) + bs_2(t)$$

$$\frac{ds_2(t)}{dt} = cs_1(t) + ds_2(t)$$

The system will blow up when the Jacobian of the system,

$$J = \begin{bmatrix} a & b \\ c & d \end{bmatrix},$$

has eigenvalues with positive real parts. That means that if one solves the system using the forward Euler method with a step size $h$, then the system with the update rule

$$\begin{pmatrix} s_1(t+kh) \\ s_2(t+kh) \end{pmatrix} = (hJ + I)^k \begin{pmatrix} s_1(t) \\ s_2(t) \end{pmatrix}$$

would blow up as $k$ increases. Let us then consider the two eigenvalues of $J$:

$$\lambda_\pm(J) = \frac{1}{2}\left(a + d \pm \sqrt{(a-d)^2 + 4bc}\right).$$

If $bc > \max(0, ad)$, then $\lambda_+(J)$ is a positive number making the system unbounded, even if $a$ and $d$ are negative.

On the other hand, suppose the system is acyclic except for self-loops (exactly what our step 1 is doing), say $ds_1(t)/dt$ depends on $s_2(t)$ but $ds_2(t)/dt$ does not depend on $s_1(t)$ anymore, then $J$ becomes

$$J = \begin{bmatrix} a & b \\ 0 & d \end{bmatrix}$$

with $a$ and $d$ as its eigenvalues. Then, the system is allowed to freely model how $s_2(t)$ affects $ds_1(t)/dt$ without concerns of explosion (no constraint on $b$), as long as $a$ and $d$ are negative.

## A8.2 Exploding gradient

Now suppose we discretize the system and solve it with RNNs, then the gradient will also behave like $(J + I)^k$, which will cause the RNNs to have exploding gradients when $k$ is large.

## A8.3 Stiffness

It is also well-known in both the physics and the neural ODE community that stiffness of the system can also cause numerical instability (Kim et al., 2021; Worsham & Kalita, 2025).

Consider the dissipative version of the 2-state system with a circular dependence:

$$\frac{ds_1(t)}{dt} = -s_1(t) + bs_2(t)$$

$$\frac{ds_2(t)}{dt} = cs_1(t) - s_2(t)$$

The eigenvalues of the system are

$$\lambda_\pm(J) = -1 \pm \sqrt{bc}.$$

If $0 < bc < 1$, then $\lambda_\pm < 0$ and the system blowing-up/exploding gradient are under control. However, the system can still suffer from stiffness, which is often defined as the ratio between the magnitude of the fastest to slowest stable rates:

$$\kappa \equiv \frac{|\lambda_-|}{|\lambda_+|} = \frac{1 + \sqrt{bc}}{1 - \sqrt{bc}}.$$

Table 4: Black-box comparison (mean $\pm$ standard error over 40 trials) (Sample Size $N = 100$)

| Model | RMSE | Peak RMSE | ENP |
|---|---|---|---|
| *True Sparsity* | | | |
| MNODE | **0.1039 $\pm$ 0.0013** | **0.4124** | 757.55 |
| BNODE | 0.1237 $\pm$ 0.0021 | 1.2451 | 3686.78 |
| S4D | 0.1130 $\pm$ 0.0012 | 0.8247 | **452.78** |
| TCN | 0.1214 $\pm$ 0.0049 | 1.8787 | 13671.23 |
| Transformer | 0.1699 $\pm$ 0.0039 | 1.2015 | 10892.65 |
| LSTM | 0.1191 $\pm$ 0.0021 | 0.9048 | 8510.78 |
| *Quasi Sparsity* | | | |
| MNODE | **0.1056 $\pm$ 0.0014** | **0.4971** | 760.73 |
| BNODE | 0.1249 $\pm$ 0.0021 | 0.7979 | 3441.73 |
| S4D | 0.1137 $\pm$ 0.0012 | 0.6873 | **456.38** |
| TCN | 0.1274 $\pm$ 0.0056 | 2.4566 | 13154.93 |
| Transformer | 0.1698 $\pm$ 0.0039 | 1.1801 | 10929.17 |
| LSTM | 0.1194 $\pm$ 0.0021 | 0.8659 | 8338.50 |

Table 5: Black-box comparison (mean $\pm$ standard error over 40 trials) (Sample Size $N = 1000$)

| Model | RMSE | Peak RMSE | ENP |
|---|---|---|---|
| *True Sparsity* | | | |
| MNODE | 0.0981 $\pm$ 0.0010 | **0.2369** | 730.25 |
| BNODE | 0.1009 $\pm$ 0.0011 | 0.5123 | 2219.60 |
| S4D | 0.0986 $\pm$ 0.0010 | 0.5624 | **602.63** |
| TCN | **0.0875 $\pm$ 0.0010** | 0.7938 | 14360.29 |
| Transformer | 0.2437 $\pm$ 0.0066 | 1.2028 | 10696.88 |
| LSTM | 0.1003 $\pm$ 0.0011 | 0.7060 | 8982.25 |
| *Quasi Sparsity* | | | |
| MNODE | 0.0987 $\pm$ 0.0010 | **0.2377** | 727.25 |
| BNODE | 0.1014 $\pm$ 0.0011 | 0.5887 | 2330.20 |
| S4D | 0.0991 $\pm$ 0.0010 | 0.4444 | **647.00** |
| TCN | **0.0896 $\pm$ 0.0011** | 0.6814 | 14173.83 |
| Transformer | 0.1420 $\pm$ 0.0027 | 0.8703 | 10576.15 |
| LSTM | 0.1008 $\pm$ 0.0011 | 0.6457 | 8981.73 |

Therefore, the system stiffness can still blow up, if $bc \uparrow 1$. On the other hand, without circular dependence, both eigenvalues are $-1$ and the stiffness is always 1 regardless of the value of $b$.

In summary, to ensure the stability of the ODE system, more complex constraints on model parameters are needed for a system with cycles than for a system without cycles.

## A9    TABULATED RESULTS

### A9.1    SYNTHETIC DATA EXPERIMENTS

We attach the tabulated version of experiment results on synthetic data in Table 4 to Table 9.

### A9.2    REAL-WORLD DATA EXPERIMENTS

We attach the tabulated version of experiment results on real-world data in Table 10 and Table 11

Table 6: Reduction method comparison (mean ± standard error over 40 trials) (Sample Size $N = 100$) Refined Initial Graph.

| Model | RMSE | Peak RMSE | ENP |
|-------|------|-----------|-----|
| GL | **0.1039 ± 0.0013** | **0.3124** | **757.55** |
| EGL | 0.1225 ± 0.0021 | 0.4826 | 1771.40 |
| NS | 0.1070 ± 0.0014 | 0.3833 | 1046.95 |
| EN | 0.1225 ± 0.0021 | 0.4820 | 1770.38 |
| RD | 0.1098 ± 0.0016 | 0.6919 | 1208.05 |
| GD | 0.1114 ± 0.0016 | 0.7097 | 1425.48 |
| NR | 0.1233 ± 0.0020 | 0.6476 | 1643.83 |

Table 7: Reduction method comparison (mean ± standard error over 40 trials) (Sample Size $N = 1000$) Refined Initial Graph.

| Model | RMSE | Peak RMSE | ENP |
|-------|------|-----------|-----|
| GL | **0.0987 ± 0.0010** | **0.2377** | **727.25** |
| EGL | 0.1058 ± 0.0013 | 0.3055 | 1772.03 |
| NS | 0.1045 ± 0.0012 | 0.3492 | 1034.40 |
| EN | 0.1066 ± 0.0013 | 0.3101 | 1771.90 |
| RD | 0.1038 ± 0.0012 | 0.3423 | 1154.30 |
| GD | 0.1040 ± 0.0011 | 0.5046 | 1420.48 |
| NR | 0.1078 ± 0.0013 | 0.3805 | 1641.48 |

Table 8: Reduction method comparison (mean ± standard error over 40 trials) (Sample Size $N = 100$) Comprehensive graph.

| Model | RMSE | Peak RMSE | ENP |
|-------|------|-----------|-----|
| GL | **0.1040 ± 0.0013** | **0.3892** | **1468.25** |
| EGL | 0.1415 ± 0.0031 | 0.8537 | 5098.98 |
| NS | 0.1845 ± 0.0039 | 1.1209 | 3565.75 |
| EN | 0.1794 ± 0.0050 | 0.9047 | 5094.78 |
| RD | 0.1236 ± 0.0022 | 0.5097 | 3905.25 |
| GD | 0.3129 ± 0.0063 | 1.8013 | 5053.78 |
| NR | 0.1235 ± 0.0020 | 0.4509 | 4933.95 |

Table 9: Reduction method comparison (mean ± standard error over 400 trials) (Sample Size $N = 1000$) Comprehensive graph.

| Model | RMSE | Peak RMSE | ENP |
|-------|------|-----------|-----|
| GL | **0.0992 ± 0.0010** | **0.2339** | **1415.03** |
| EGL | 0.1287 ± 0.0022 | 0.5938 | 5097.38 |
| NS | 0.1758 ± 0.0035 | 0.6184 | 3881.53 |
| EN | 0.1287 ± 0.0022 | 0.5397 | 5095.23 |
| RD | 0.1278 ± 0.0021 | 0.4183 | 4410.73 |
| GD | 0.3415 ± 0.0058 | 1.5982 | 5048.73 |
| NR | 0.1165 ± 0.0017 | 0.3507 | 4946.93 |

Table 10: Predictive performance (mean $\pm$ standard error over 10 trials).

| Model | RMSE | MAPE | Corr | Acc. |
|---|---|---|---|---|
| MNODE_NR | $36.19 \pm 0.33$ | $0.230 \pm 0.002$ | $0.649 \pm 0.006$ | $0.760 \pm 0.004$ |
| MNODE_DK | $36.58 \pm 0.60$ | $0.229 \pm 0.003$ | $0.657 \pm 0.006$ | $0.765 \pm 0.004$ |
| MNODE_HGS12 | $35.92 \pm 0.31$ | $0.227 \pm 0.002$ | $0.657 \pm 0.004$ | $0.768 \pm 0.002$ |
| MNODE_HGS1 | $36.12 \pm 0.40$ | $0.229 \pm 0.003$ | $0.648 \pm 0.005$ | $0.764 \pm 0.005$ |
| MNODE_HGS2 | $35.96 \pm 0.30$ | $0.229 \pm 0.002$ | $0.650 \pm 0.005$ | $0.766 \pm 0.002$ |
| MNODE_HGS | $\mathbf{35.22} \pm 0.25$ | $\mathbf{0.223} \pm 0.002$ | $\mathbf{0.682} \pm 0.003$ | $\mathbf{0.786} \pm 0.002$ |
| MNODE_HGS3 | $36.13 \pm 0.41$ | $0.229 \pm 0.002$ | $0.651 \pm 0.005$ | $0.768 \pm 0.003$ |
| MNODE_HGS13 | $35.82 \pm 0.37$ | $0.227 \pm 0.002$ | $0.654 \pm 0.007$ | $0.773 \pm 0.004$ |
| MNODE_HGS23 | $35.95 \pm 0.27$ | $0.228 \pm 0.002$ | $0.669 \pm 0.003$ | $0.769 \pm 0.003$ |
| MNODE_EGL | $35.86 \pm 0.25$ | $0.227 \pm 0.001$ | $0.650 \pm 0.005$ | $0.760 \pm 0.003$ |
| MNODE_EN | $35.93 \pm 0.29$ | $0.227 \pm 0.002$ | $0.651 \pm 0.005$ | $0.760 \pm 0.002$ |
| MNODE_NS | $35.55 \pm 0.28$ | $0.227 \pm 0.002$ | $0.655 \pm 0.005$ | $0.768 \pm 0.003$ |
| MNODE_GD | $36.70 \pm 0.44$ | $0.229 \pm 0.002$ | $0.655 \pm 0.007$ | $0.765 \pm 0.003$ |
| MNODE_RD | $35.78 \pm 0.41$ | $0.227 \pm 0.003$ | $0.662 \pm 0.002$ | $0.769 \pm 0.003$ |
| BNODE | $37.08 \pm 0.25$ | $0.260 \pm 0.002$ | $0.666 \pm 0.003$ | $0.759 \pm 0.003$ |
| S4D | $42.91 \pm 0.39$ | $0.283 \pm 0.003$ | $0.629 \pm 0.005$ | $0.724 \pm 0.002$ |
| LSTM | $40.69 \pm 0.39$ | $0.266 \pm 0.003$ | $0.666 \pm 0.003$ | $0.733 \pm 0.003$ |
| TCN | $41.09 \pm 0.44$ | $0.277 \pm 0.003$ | $0.672 \pm 0.003$ | $0.725 \pm 0.002$ |
| Transformer | $46.29 \pm 0.51$ | $0.283 \pm 0.003$ | $0.592 \pm 0.004$ | $0.664 \pm 0.009$ |

Table 11: Complexity and peak-error metrics. Variance shows mean $\pm$ standard error over 10 trials; ENP and peak metrics are means only.

| Model | Variance | ENP | Peak RMSE | Peak MAPE |
|---|---|---|---|---|
| MNODE_NR | $125.8 \pm 17.9$ | 10684 | 189.6 | 1.401 |
| MNODE_DK | $149.2 \pm 34.3$ | 6956 | 183.3 | 1.464 |
| MNODE_HGS12 | $119.9 \pm 17.1$ | 9033 | 202.7 | 1.469 |
| MNODE_HGS1 | $136.1 \pm 21.5$ | 8848 | 177.9 | 1.398 |
| MNODE_HGS2 | $115.4 \pm 20.5$ | 10643 | 193.7 | 1.426 |
| MNODE_HGS | $\mathbf{76.4} \pm 17.9$ | **7551** | **123.4** | **1.222** |
| MNODE_HGS3 | $101.5 \pm 23.8$ | 7966 | 167.8 | 1.313 |
| MNODE_HGS13 | $96.6 \pm 17.2$ | 7735 | 169.3 | 1.354 |
| MNODE_HGS23 | $96.4 \pm 19.9$ | 8054 | 166.6 | 1.354 |
| MNODE_EGL | $92.0 \pm 12.4$ | 8326 | 169.4 | 1.321 |
| MNODE_EN | $101.7 \pm 19.0$ | 8548 | 169.0 | 1.239 |
| MNODE_NS | $96.8 \pm 21.5$ | 8730 | 161.3 | 1.330 |
| MNODE_GD | $124.7 \pm 23.8$ | 8861 | 260.6 | 1.349 |
| MNODE_RD | $107.0 \pm 22.4$ | 8955 | 184.8 | 1.322 |
| BNODE | $332.5 \pm 17.4$ | 8596 | 190.9 | 1.541 |
| S4D | $291.4 \pm 11.3$ | 8099 | 194.3 | 1.667 |
| LSTM | $317.8 \pm 27.9$ | 8102 | 178.6 | 1.437 |
| TCN | $384.5 \pm 23.4$ | 8261 | 161.6 | 1.803 |
| Transformer | $509.0 \pm 27.8$ | 8122 | 210.2 | 1.953 |

