# OpenReview forum: "Automatic and Structure-Aware Sparsification of Hybrid Neural ODEs with Application to Glucose Prediction"
_ICLR.cc/2026/Conference — ICLR 2026 Poster_

### Official Review · Reviewer_5nbh · 2025-10-20

**Soundness:** 1
**Presentation:** 3
**Contribution:** 2
**Rating:** 4
**Confidence:** 4

**Summary:**

This paper presents HGS, a method for simplifying the structure of mechanistic neural ODEs. The authors point out that the mechanistic models often used may be too complex for the small datasets we typically have in fields like healthcare, which can lead to overfitting. Their proposed solution includes three steps: first to modify the model's graph structure based on its topology by collapsing cycles and adding shortcuts, and then to use regularization to learn which connections to prune. The method is evaluated on synthetic data and a real world glucose forecasting task.

**Strengths:**

1. The work is well motivated. Trying to bridge the gap between complex mechanistic models built by experts and data-driven methods is a critical research area, and the paper does a good job of framing the problem.
2. The idea to first perform a structural simplification of the graph before applying a data-driven pruning is quite clever. It's a nice way to inject some domain agnostic heuristics to constrain the learning problem.
3. The set of experiments seems comprehensive. The authors have benchmarked their method against a good range of strong baselines, and the ablation study clearly shows that each part of their pipeline contributes to the final result.

**Weaknesses:**

My main concerns are with the real-world application, and I would need the authors to address these points before I could reconsider my score.

1. My primary concern is how the model handles patient variability. The T1DEXI dataset includes 105 different people, and it's a physiological fact that glucose dynamics differ significantly between individuals. From my reading of Equation 1, the MNODE learns a single set of dynamics for everyone, i.e., an "average patient" model. This seems to sidestep an important challenge in this domain. Could you clarify if this is indeed the case, and if so, explain the rationale behind this choice? To be clear, this has been discussed extensively in recent years ([1-4] for example).
2. The paper mentions that the cross validation splits were created from a "random permutation" of the 342 time series. To avoid data leakage, it's standard practice in clinical ML to split data at the patient level (i.e., all data from a single person stays in one fold). Could you confirm whether your evaluation followed this practice? If not, the reported performance might not accurately reflect generalization to new, unseen patients.
3. The paper uses the term MNODE, but it seems that the functional forms of the UVA-Padova model are discarded, with only the graph structure being retained. The actual dynamics are then learned by MLPs. This is a very weak form of mechanistic prior. It would be helpful to discuss the trade-offs here.
4. It's worth noting that in the synthetic experiments, the TCN model starts to outperform HGS on the RMSE metric as the sample size grows to N=1000. While HGS remains more robust in terms of Peak RMSE, this suggests that its strong inductive bias might become a disadvantage when more data is available. Could you comment on this trade-off?

[1] Generative ODE Modeling with Known Unknowns

[2] Physics-Integrated Variational Autoencoders for Robust and Interpretable Generative Modeling

[3] Learning Physics Constrained Dynamics Using Autoencoders

[4] CONFIDE: Contextual Finite Difference Modelling of PDEs

**Questions:**

Please address the weaknesses above.

---

> ### Author Response · Authors · 2025-11-20
>
> **My primary concern is how the model handles patient variability. ... Could you clarify if this is indeed the case, and if so, explain the rationale behind this choice? To be clear, this has been discussed extensively in recent years ([1-4] for example).**
>
> First, we want to emphasize that our proposed model is also tailored for individual patient in the sense that the initial values of relevant latent states of the individual is determined from the 4-hour history of that patient via an encoder, which can capture all the inter- and intra- patient variabilities at that instance. Those instance-specific initial values affect neural ODE solutions as much as ODE system parameters, which are indeed the same across instances.
>
> Lastly, our method can be used to train the prediction model for individual patient if there is sufficient number of instances (data) for the specific patient of interest. In our applications, we have a large number of patients but only a few usable instances per patient, so can't train different prediction models for different patients. In clinical practice, the prediction model will often be used to make prediction for new patients. In such a case, prediction models specifically trained for existing patients (even if feasible) would not be useful. On the other hand, our prediction model is trained to optimize the average performance in a population and should have a good prediction accuracy for new patients from the same population.
>
> In the revised submission, we have modified equation (1) to more explicitly reflect instance-dependency of the initial condition.
>
>
>
> **The paper mentions that the cross validation splits were created from a ``random permutation'' of the 342 time series. To avoid data leakage, it's standard practice in clinical ML to split data at the patient level (i.e., all data from a single person stays in one fold). Could you confirm whether your evaluation followed this practice? If not, the reported performance might not accurately reflect generalization to new, unseen patients.**
>
> Thanks for the insightful comments.  We are aware of the risk of data leakage in evaluating the model performance.  However, our particular application is special:  it is well recognized that intra-patient variability can be as high as inter-patient variability among diabetic patients [1-3].  Therefore, predicting new cases of an existing patient is as hard as new patients.
> In our dataset, the average intra-patient and inter-patient variabilities (measured in terms of root-mean-squared-difference) of average glucose level are 54.24 and 55.10 mg/dl, respectively, also supporting the aforementioned claim.
>
> Another important reason that we didn't conduct patient-level cross-validation is that we expect to deploy the model on both existing patients and new patients in practice. Thus, the model performance needs to evaluated on new cases of existing patients as well as on new patients. Splitting at patient level would overlook the former.
>
> We have included a sentence in experiment section (page 8) and an extra paragraph in Appendix 4 to highlight this.
>
>
> [1] Identification of intra-patient variability in the postprandial response of patients with type 1 diabetes, Laguna et al, 2014
>
> [2] Intra-patient dynamic variations in Type 1 Diabetes: A review, Moscoso-Vásquez et al, 2016
>
> [3] Substantial Intra-Individual Variability in Post-Prandial Time to Peak in Controlled and Free-Living Conditions in Children with Type 1 Diabetes, Bell et al, 2021

---

> ### Author Response · Authors · 2025-11-20
>
> **The paper uses the term MNODE, but it seems that the functional forms of the UVA-Padova model are discarded, with only the graph structure being retained. The actual dynamics are then learned by MLPs. This is a very weak form of mechanistic prior. It would be helpful to discuss the trade-offs here.**
>
> We agree that the functional forms of UVA-Padova model are not used in the current proposal. However, several versions of hybrid modeling utilizing the functional forms are possible:
> - keep the functional form and only tune the ODE parameter $\beta(t)$ in
> $\frac{ds(t)}{dt}=f_m\left(s(t),t;\beta(t)\right) ~\mbox{ and } \beta(t)=NN(s(t),t)$
> - keep functional form and add an NN to learn residuals: $\frac{ds(t)}{dt}=f_m(s(t),t; \beta)+NN(s(t),t)$
> - completely discard the functional form: $\frac{ds(t)}{dt}=NN(s(t), t)$
>
> These approaches and their trade-offs have been examined in [4]. In general,
> - more mechanistic prior: more mechanistic constraint on model space, require less training data, less likely to overfit, but less capable of learning complex/unknown dynamics not captured by the prior.
> - more black-box: less constraint on model space, more flexible and capable of learning new dynamics/complex patterns, but more prone to over-fitting and require more training data to learn a good model.
>
> We have added a paragraph to related work to discuss general hybrid modeling techniques and trade-offs, For a more comprehensive discussion of hybrid modeling approach, please see [4].
>
> **It's worth noting that in the synthetic experiments, the TCN model starts to outperform HGS on the RMSE ... While HGS remains more robust in terms of Peak RMSE ... Could you comment on this trade-off?**
>
> Thanks for the insightful comments. Indeed, while informative, the mechanistic prior is never perfect.  While it brings down the variance, it also may induce systematic bias.  When sufficient data are available for training, the role of bias would dominate that of the variance in determining the predictive performance, and black-box models may be superior as our simulation study demonstrated.
>
> [4] Hybrid2 Neural ODE Causal Modeling and an Application to Glycemic Response, Zou et al, 2024

---

> > ### Comment · Reviewer_5nbh · 2025-11-24
> >
> > Thank you for taking the detailed rebuttal.
> >
> > **Re patient heterogeneity:**
> > I appreciate the clarification regarding the role of the encoder and the initial conditions in capturing patient variability. While I maintain that explicitly conditioning the dynamics function on static patient features (demographics, clinical parameters) would be a more robust way to handle physiological heterogeneity than relying solely on the initial latent state, I acknowledge that your approach is a valid design choice in the context of latent neural ODEs, especially given the data constraints. The modification to Equation 1 helps clarify this dependency.
> >
> > **Re cross-validation and data leakage:**
> > The statistical evidence you provided regarding the similarity between intra-patient and inter-patient variability in this specific dataset (RMSD 54.24 vs 55.10) effectively addresses my concern about data leakage. It suggests that "learning the patient" does not provide the trivial advantage it might in other clinical domains.
> >
> > Consequently, I am raising my score to 6.

---

> > > ### Author Response · Authors · 2025-11-24
> > >
> > > Thank you very much for taking the time to read our rebuttal and update the score! We are glad that we were able to address your initial concerns, and we appreciate all the valuable feedback you provided to help us improve the paper.

---

### Official Review · Reviewer_egCh · 2025-10-24

**Soundness:** 3
**Presentation:** 2
**Contribution:** 2
**Rating:** 6
**Confidence:** 3

**Summary:**

The authors propose a sparsification method for Neural ODEs parameterized by DAGs / relaxed DAGs, and benchmark on predictive accuracy for held-out samples from time-series data.

**Strengths:**

The authors get good results on their synthetic and empirical benchmarks, and their graph reduction seems like a reasonable solution to the problem.  The comparison to other graph reduction methods makes the argument in favor of the current setup more convincing.

**Weaknesses:**

I’m a little confused by the motivation of the setup.  In most instances I’m familiar with, the incentive for parameterizing a causal model or a dynamical system with a graph is interpretability, but the authors insist that the setup isn’t identifiable and the purpose is only for predicting future states.  In that case, what is the rationale for insisting on DAG / RDAG structure at all?  One could still encode prior knowledge by constraining dependence of one state variable on others with masking.

The third part of the model seems to just be the inclusion of L1/L2 regularization, which I think is straightforward enough to not be a novel portion of the proposed method.

The highly varying y-axes across graphs makes it much much harder to gauge which performance improvements are substantial and which ones are almost negligible.  I would suggest the authors use tables in some cases so a reader can more easily tell.  For example, many bar plots in Figure 2b are somewhat misleading because the scale is so small.

**Questions:**

To my understanding, only the MNODE based methods are given the prior graph determining the underlying dynamics?  In that case it seems the comparisons to the black box NODE models are somewhat uninformative, without encoding that prior into the model (even as a form of regularization).

---

> ### Author Response · Authors · 2025-11-20
>
> **I’m a little confused by the motivation of the setup. In most instances I’m familiar with, the incentive for parameterizing a causal model or a dynamical system with a graph is interpretability, but the authors insist that the setup isn’t identifiable and the purpose is only for predicting future states. In that case, what is the rationale for insisting on DAG / RDAG structure at all? One could still encode prior knowledge by constraining dependence of one state variable on others with masking.**
>
> Thanks for raising this important question. The interpretability is indeed important and is an advantage of a mechanistic model. We stated explicitly in intro and related work that our method has better interpretability than black-box methods. The reduced mechanistic model can be used for hypothesis generating and used for prediction purpose. Our caution of non-identifiability is about the risk of   causally interpreting of the original or reduced graph of the mechanistic model. Specifically, one should not consider a directional edge in the graph suggests that the value in one state determines the value in another state. Such a causal relationship requires validation from randomized experiments to perturb related states while controlling all potential confounders.
>
> We have modified section 2.5 to reflect this and included a paragraph to discuss mechanistic interpretations of our results.
>
> Moreover, graph formalism provides a convenient and rigorous way to articulate our proposed modification to the original mechanistic model. Without terminologies such as maximal strongly connected components and transitive closures, it would be very cumbersome to rigorously define intuitive notions of "circular dependence" ($v_1 \rightarrow v_2 \cdots \rightarrow v_K \rightarrow v_1$) and ``short-cuts'' ($v_1 \rightarrow v_2 \rightarrow v_3 \Rightarrow v_1 \rightarrow v_3$). It also allows one to use out-of-shelf graph algorithms to implement step 1 and step 2 modifications rather than manually adding/masking states.
>
>
> **The third part of the model seems to just be the inclusion of L1/L2 regularization, which I think is straightforward enough to not be a novel portion of the proposed method.**
>
>  We agree that the regularization idea is very natural and non-surprising. However, unlike well established regularizations such as L2 or L1 on the NN parameters, we believe that the effect of coupling L1 regularization on edge specific weights with L2 regularization on neural network parameters is not well-understood in ML community.  We have established the connection of this hybrid L1/L2 regularization with the conventional group LASSO penalty on the NN parameters, which has been studied extensively in statistics literature. The new formulation is more general than the group lasso regularization and allows an efficient optimization algorithm. Therefore, we consider this proposal a novel contribution to the field.
>
>
> **The highly varying y-axes across graphs makes it much much harder to gauge which performance improvements are substantial and which ones are almost negligible. I would suggest the authors use tables in some cases so a reader can more easily tell. For example, many bar plots in Figure 2b are somewhat misleading because the scale is so small.**
>
> Thanks for pointing this out. It is indeed difficult to gauge the performance improvement, as scale and magnitude of these metrics are highly domain and data dependent.  To address your concern, we provided a table version of the results in the Appendix 9 for interested reader, while keeping the Figures as a simple visualization of the presence or absence of difference in performance between different methods.
>
> **To my understanding, only the MNODE based methods are given the prior graph determining the underlying dynamics? In that case it seems the comparisons to the black box NODE models are somewhat uninformative, without encoding that prior into the model (even as a form of regularization).**
>
> We agree that the black-box models don't incorporate any prior domain knowledge. What we explore here is the trade-off between making the learning process fully data-driven and incorporating useful but potentially imperfect inductive bias. As another reviewer suggested, readers may be interested in quantifying such trade-offs under different data regimes.  As shown by our synthetic experiments, it is not true that having a mechanistic prior will always be better, as the mechanistic prior, which in medicine is often reasonably good but not as accurate as in physics, may miss real-world complexities and restrict the prediction model. While it brings down the variance, it also may induce systematic bias.  When sufficient data are available for training, the role of bias would dominate that of the variance in determining the predictive performance, and black-box models may be superior as our simulation study demonstrated.
>
> We have included relevant discussions at the end of Section 3.

---

> > ### Comment · Reviewer_egCh · 2025-11-25
> >
> > I appreciate the authors comments, they have addressed my concerns, especially the section justifying the RDAG structure as a way to regularize the underlying ODEs.  I will keep my positive score.

---

> > > ### Author Response · Authors · 2025-11-25
> > >
> > > Thank you for responding to our rebuttal and keeping the positive score!

---

### Official Review · Reviewer_mjPg · 2025-10-31

**Soundness:** 3
**Presentation:** 2
**Contribution:** 2
**Rating:** 4
**Confidence:** 3

**Summary:**

The paper targets an application of Hybrid ODE models in healthcare. Hybrid neural ODEs combine neural ODEs with mechanistic models to incorporate domain relational knowledge. However, mechanistic models in physiology and medicin tend to become excessively large when capturing wide ranging dynamics.  These models may contain  many latent states for a handful of observable states, and interactions among states can lead to inefficiency and overfitting. Model reduction techniques often require domain knowledge. Data-driven reduction techniques, on the other hand, are completely devoid of any domain knowledge and do not preserve mechanistic structure or constraints.

The paper proposes a method for automatic state selection and structure optimization in mechanistic neural ODEs by combining graph modification based on domain knowledge with data-driven regularization that improve predictive performance and stability while remaining mechanistically plausible. The approach combines domain-knowledge informed graph modification with a mix of L1 and L2 regularization . The graph modification bases itself on classical reduction methods to retain key topological structures. While the regularization step allows gradient based pruning during training, making the processes efficient.

The architecture of the model is an encoder-decoder on where the encoder takes in history and produces an initial state. The decoder takes the initial state and graph representation of the ODE system and future inputs evolves the state features as a Neural ODE.

The actual reduction algorithm has three steps. Step 1 merges maximally strongly connected components into supernodes with self loops. Step 2 augments the graph with short-cuts using partial transitive closure, and step 3 applies a data-driven mixture of L1 and L2 regularization.

Experiments are performed on healthcare data showing improved prediction and robustness with desired sparsity.

**Strengths:**

The paper is very detailed regarding the method and well organized.

The method combines classical and data-driven approaches for the state reduction and sparsification problem.

Experiments are done both on synthetic and real data. Ablation experiments are done for all the steps of the method.

The paper targets biomedical domains where methods must work with complex models and limited data and interpretability is significant.

**Weaknesses:**

Rationale of step 1 says that transforming the original graph into and RDAG reveals essential causal structure. What is the evidence for this? It also says that this improves training stability. Has this been shown in the experiments? What is the reason for this improvement in stability?

How do self-loops allow more flexible modeling of intra-component dynamics? It is said to be a key principle of hybrid modeling (line 190), but I am unsure how this is established.

The formulation of step 2 is too formal which leads to a decrease in clarity of the paper. It would be useful to start from intuition and build from there. For instance, it would be useful to first describe what is the intuitive description of a partial closure.

The introduction to the paper claims that the method allows incorporating domain knowledge reduction of the graph. As far as I can tell in steps 1 and 2, the graph reduction methods are more structural (not data-driven) rather than incorporating domain knowledge.

Typos etc.

line 167. “maximally strongly” should be maximal strongly …
There appears to be a formatting problem where line spacing has changed starting from page 5.

**Questions:**

In the motivation of the method the paper states that the method allows domain-knowledge informed graph modification. I don’t follow how the method allows this in the steps of the method. How is this domain-knowledge used in the experiments?

---

> ### Author Response · Authors · 2025-11-20
>
> **Rationale of step 1 says that transforming the original graph into and RDAG reveals essential causal structure. What is the evidence for this?**
>
> A clean causal interpretation does not permit statements such as A causes B and B causes A simultaneously. Therefore, it is standard practice to transform graphs containing cycles into DAGs or restricted DAGs (RDAGs) to explore plausible causal structures. By treating each cycle as a single unit within the system, the resulting DAG yields an interpretable and simplified causal representation that preserves the essential elements of the original mechanistic system.
>
> **It also says that this improves training stability. Has this been shown in the experiments? What is the reason for this improvement in stability?**
>
> ODE systems and neural ODEs whose dependency graphs contain cycles are susceptible to three major sources of instability: blow-ups, exploding gradients, and stiffness. While blow-ups and exploding gradients are familiar concepts in the machine-learning community, stiffness is less commonly discussed. Nonetheless, stiffness is a critical property of ODE systems because stiff neural ODEs can be extremely unstable and difficult to solve numerically. Please see references (Stiff neural ordinary differential equations, Kim et al, 2021, A guide to neural ordinary differential equations: Machine learning for data-driven digital engineering, Worsham and Kalita, 2025.)
>
> The following two examples are added to the Appendix to illustrate these instabilities in a two-stage system with cycles.  Consider a simple 2 state system with a circular dependence:
> $$\frac{ds_1(t)}{dt}=as_1(t)+bs_2(t)$$
> $$\frac{ds_2(t)}{dt}=cs_1(t)+ds_2(t)$$
>
> Let $\lambda_1$ and $\lambda_2$ be the eigenvalues of the Jacobian of the system.
> If $\lambda_i, i=1$ or 2 has a large positive real part, then $|h\lambda_i+1|>1$ and the system is unstable in that if one solves the ODE system using forward Euler method with a step size $h,$ then the update
> $$S(t+kh)=\left(hJ+I\right)^kS(t)$$
> blows up as $k$ increases. The two eigenvalues of $J$ are
> $$\frac{1}{2}\left(a+d\pm\sqrt{(a-d)^2+4bc}\right).$$
> If $bc> \max(ad, 0)$, the larger eigenvalue has a positive real part and the system is unstable. On the other hand, suppose the system is acyclic except for self-loops, say $ds_1(t)/dt$ depends on $s_2(t)$ but $ds_2(t)/dt$ does not depend on $s_1(t)$, then $c=0$ and the eigenvalues of the upper triangular Jacobian are simply $a$ and $d$.  In such a case, this system is allowed to freely chose $b$ to model how $s_2(t)$ affects $ds_1(t)/dt$ without concerns of explosion, as long as $\max(a, d)<0$.  Similarly, when we discretize the ODE system and solve it using RNNs with residual connections, the gradient will also behave like $\left(J+I\right)^k,$ which will cause the RNNs to have exploding gradients for large $k$. In summary, in the presence of cycles, there are more constraints on model parameters to ensure the stability of the system.
>
> Second, consider the following dissipative version of the 2-state system with a circular dependence to illustrate the system stiffness:
> $$\frac{ds_1(t)}{dt}=-s_1(t)+bs_2(t)$$
> $$\frac{ds_2(t)}{dt}=cs_1(t)-s_2(t),$$
> with eigenvalues $-1\pm\sqrt{bc}$. Suppose that we have controlled for blowing up, e.g., $0<bc<1.$ The system stiffness is defined as the ratio between the magnitude of the fastest to slowest stable rates:
> $$\kappa \equiv \frac{1+\sqrt{bc}}{1-\sqrt{bc}}.$$
> Thus, $\kappa$ may still go to infinity, when $\sqrt{bc}\uparrow 1.$
> On he other hand, without circular dependence (i.e., $c=0$,) these two eigenvalues are both $-1$ and the stiffness is a constant regardless of the value of $b.$  Again, there are more constraints to avoid the system stiffness in the presence of cycles.
>
> Finally, our ablation study also confirms the benefits of this approach: the full HGS model consistently outperforms the version without Step 1 in both predictive accuracy and robustness.
>
> **How do self-loops allow more flexible modeling of intra-component dynamics? It is said to be a key principle of hybrid modeling (line 190), but I am unsure how this is established.**
>
> By collapsing cycles into super-nodes, we reduce the risk of system instability by relaxing the constraints placed on the model parameters. To preserve the postulated dynamics within each collapsed cycle, we incorporate a self-loop with a flexible neural network at the corresponding super-node, compensating for the structural simplification. The key principle, as stated in the pioneering work of Physics Informed Neural Networks by Raissi, Perdikaris and Karniadakis, is that NNs can approximate solutions to dynamical systems well.
>
> We have re-written the rationale of step 1 in the main text to be more precise about what we mean and also added two toy examples in Appendix 8 to support the claim.

---

> ### Author Response · Authors · 2025-11-20
>
> **The formulation of step 2 is too formal which leads to a decrease in clarity of the paper. It would be useful to start from intuition and build from there. For instance, it would be useful to first describe what is the intuitive description of a partial closure.**
>
> Intuitively, one may think of a physiological path as a student’s high-school journey moving through grades 9 to 12. Normally, the student progresses step by step—9 → 10 → 11 → 12. A **transitive closure** adds all possible “skip-grade” links, letting the student jump directly from grade 9 to 11 or 12, or from 10 to 12, as long as they always move to a higher grade (obey the reachability relations). A **partial transitive closure** is a more cautious version: it allows some skipping but forbids overly aggressive jumps, like going straight from 9 to 12. The idea is that, just as students progress at different speeds, biological processes/pathways in physiological systems also vary in how many intermediate states they pass through and can therefore often be better modeled with fewer latent states.  For example, quasi-steady-state approximations in chemical kinetics eliminate fast variables by assuming equilibrium. By adding shortcuts (transitive closure), the model gains flexibility to capture these differences without discarding realistic reachability constraints. In addition, by enforcing partial transitive closure, we block direct input–output connections that bypass all latent states, enabling the model to represent the lagging effects that are common in biomedical research.
>
> We have included this paragraph in rationale of step 2.
>
> **The introduction to the paper claims that the method allows incorporating domain knowledge reduction of the graph. As far as I can tell in steps 1 and 2, the graph reduction methods are more structural (not data-driven) rather than incorporating domain knowledge.**
>
> Thanks for the constructive comment. In the original paper, we have tailored our algorithm to general biomedical domains by suggesting
>
> - all cycles should be removed (because in physiology, feedback loops are prevalent and are not as indispensable as in physical systems);
> - all short-cuts are permissible except those bypassing all latent states (because physiological models rarely model the response directly as a function of input);
> - all edges should be regularized (because there is often no strong clinical evidence about the necessity of particular edges).
>
> In the revised submission, we have clarified that those are general suggestions for biomedical applications and are not rigid rules: users can still input their domain knowledge at specific applications by
>
> - setting up the mechanistic prior;
> - selecting cycles to be collapsed;
> - selecting short-cuts to be added;
> - selecting edges to be kept (i.e.,  not regularized in step 3).
>
>
> **line 167. “maximally strongly” should be maximal strongly … There appears to be a formatting problem where line spacing has changed starting from page 5.**
>
> Thanks for pointing this out, we have edited the word for MSCC. For the line spacing, we did not make modifications to the template or in the source file, it seems to be an artifact of the template itself adjusting line spacing adaptively.

---

> ### Comment · Reviewer_mjPg · 2025-11-28
>
> I appreciate the response and for including the clarifications in the submission. My main concern was clarity of the description. I will raise my score to 6.

---

> > ### Author Response · Authors · 2025-11-28
> >
> > Thank you for responding to our rebuttal and raising the score. We are glad that we are able to address your concerns.

---

### Official Review · Reviewer_Sy1g · 2025-11-01

**Soundness:** 4
**Presentation:** 3
**Contribution:** 4
**Rating:** 8
**Confidence:** 3

**Summary:**

This paper addresses a key bottleneck in hybrid neural ODEs (i.e., over-parameterization and over-fitting arising from combination of mechanistic model complexity, addition of data-driven components and sparse data) by proposing an automatic structure-learning pipeline. The method combines domain-informed graph modifications with data-driven regularization to identify essential states and interactions in mechanistic neural ODEs, thereby improving training efficiency, prediction accuracy, and robustness. The authors evaluate performance on synthetic and a real healthcare dataset, showing clear improvements over existing baselines while yielding sparser and more interpretable models. Overall, this is a strong paper. The problem is relevant and timely, the methodology is technically sound, and the experiments convincingly support the claims.

**Strengths:**

Originality: The paper addresses an important challenge in hybrid neural ODEs: automatic reduction of mechanistic state space and structure. The authors propose a creative combination of domain-guided graph pruning and regularization-driven sparsification, contributing to the emerging literature on structure-learning for scientific ML. Although I must admit I am not an expert in graph-based techniques so I am not able to assess the novelty of these claims, but the authors acknowledge a large body of literature.

Quality: The authors present a solid methodological formulation with thoughtful integration of mechanistic priors and data-driven sparsity and strong empirical validation across both synthetic and real healthcare datasets, confirming generality. Moreover, they perform a comprehensive comparison against state-of-the-art baselines.

Clarity: The problem is well-motivated, and writing is generally clear and well-structured. The authors make clear statements about what this method is and what it is not (e.g., does not guarantee true model discovery). Overall, the model architecture and training procedures are communicated effectively, multiple repetitions are performed for all results to confirm robustness and claims are aligned with results.

Significance: This paper positions hybrid modeling for more deployment-ready use in data-scarce healthcare contexts, which is a domain where interpretability and robustness are essential. In addition, automated model reduction has broad relevance beyond the presented domain and opens a promising direction for robust structure learning in hybrid models.

**Weaknesses:**

Figures readability: Many figures contain very small titles/legends/axis labels, making interpretation difficult. Increasing font sizes and improving layout will improve accessibility and impact.

Interpretation depth: While results are strong, the text describing figures and drawing insights is brief. Slightly reducing methodological narrative to provide richer interpretation would improve readability and scientific impact.

Limited discussion of learned functional forms: The paper does not clearly articulate whether the recoverable mechanistic relationships must be linear w.r.t. parameters (as in LASSO-type sparsity) or whether the framework can support nonlinear mechanistic hypotheses. Clarifying this assumption and its implications for general healthcare systems would strengthen the positioning.

Computational considerations not fully reported: Training time, scalability to larger mechanistic graphs, and computational savings from sparsification could be discussed more concretely in Appendix. Would this be a concern in the practical implementation and use in the types of healtcare applications of interest?

**Questions:**

Model class expressiveness: Does the proposed sparsification framework assume linearity in the mechanistic model parameters? If so, how does this limit applicability to nonlinear systems common in healtchare and other fields?

Generalization to larger mechanistic graphs: How does performance scale with increasing state dimensionality or interaction complexity? Any expected limitations?

Practical deployment: What guidance can you provide for practitioners on choosing sparsity regularization strengths and graph-modification thresholds?

Training efficiency: Can you quantify computational savings (e.g., reduced training time, memory use) achieved through the learned sparsity?

Interpretability: Are there examples where the discovered reduced model aligns with clinical or mechanistic knowledge? If so, including brief commentary would strengthen the healthcare relevance.

---

> ### Author Response · Authors · 2025-11-20
>
> **Model class expressiveness: Does the proposed sparsification framework assume linearity in the mechanistic model parameters? If so, how does this limit applicability to nonlinear systems common in healthcare and other fields?**
>
> We did not impose any specific functional form for the mechanistic model and our method can be used to simplify general nonlinear mechanistic models.
>
> **Generalization to larger mechanistic graphs: How does performance scale with increasing state dimensionality or interaction complexity? Any expected limitations?**
>
> First, our framework imposes no restrictions on interactions between states. Second, as demonstrated in our synthetic experiment comparing reduced versus comprehensive starting graphs, the benefit of our method becomes more pronounced when the initial graph is large and contains redundant edges. In our current work, we focus on biomedical applications, where clinically validated mechanistic models rarely involve more than 50 states because constructing and validating substantially larger models is extremely challenging in practice.
>
> We agree that extending the approach to massive graphs with hundreds or thousands of states is conceptually appealing. Based on our limited experience, such generalization appears feasible, and the associated computational burden should increase approximately linearly with the number of states. Exploring applications in domains where high-dimensional mechanistic models are common represents an important future research direction. Additional methodological developments—such as knockoff-based procedures—may be required to enable valid inference on the sparsified model.
>
> **Practical deployment: What guidance can you provide for practitioners on choosing sparsity regularization strengths and graph-modification thresholds?**
>
> We use standard cross-validation to choose these hyper-parameters.
>
> **Training efficiency: Can you quantify computational savings (e.g., reduced training time, memory use) achieved through the learned sparsity?**
>
> - Model reduction methods: HGS, NS, EGL, EN all take about 2 hours to complete
> - Greedy and Random searches for model reduction: it takes about 24 hours to complete
> - Retrain the reduced model: it takes about 1 hour to train HGS reduced model
> - Train the full model: it takes about 2 hours to train the unreduced model
>
> While these computation time would depend on the hardware environment, the relative time saving of our proposal in comparison with other methods such as greedy search is expected to be reproducible.
>
> **Interpretability: Are there examples where the discovered reduced model aligns with clinical or mechanistic knowledge? If so, including brief commentary would strengthen the healthcare relevance.**
>
> In our real-data example, the method eliminates edges associated with glucagon feedback, consistent with the well-established observation that individuals with T1D exhibit impaired glucagon responses during hypoglycemia (Hypoglycemia and Diabetes: A Report of a Workgroup of the American Diabetes Association and The Endocrine Society, Seaquist et al., 2011). Our findings further suggest that this suppression persists even during exercise-induced hypoglycemia—a novel hypothesis that may warrant future investigation.
>
> In addition, the learned graph prunes intermediate compartments linking insulin and glucose, effectively identifying a shorter regulatory pathway. This result aligns with empirical evidence that insulin sensitivity increases during and after exercise, producing a more rapid glucose response (A systematic review and meta-analysis of exercise interventions in adults with type 1 diabetes, Yardley et al., 2014).
>
> We have included a short paragraph of comment at the end of the experiment section in the revised pdf.

---

### Author Response · Authors · 2025-12-03
**Rebuttal Summary**

Dear AC,

We would like to provide the following summary of the rebuttal discussions to facilitate the meta-review process.

## **Final Score Changes Overview**

| Reviewer | Initial Score | Final Score |
|----------|---------------|-------------|
| **Sy1g** | 8 | **8** |
| **mjPg** | 4 | **6** |
| **egCh** | 6 | **6** |
| **5nbh** | 4 | **6** |

**Overall:** Two reviewers raised their scores from borderline-reject (4) to borderline-accept (6).
No reviewer lowered their score.

---

# **Reviewer-by-Reviewer Summary**

---

### **1. Reviewer Sy1g — Score: 8 → 8 (unchanged)**

### **Main Concerns**
- Need deeper interpretation of experimental results.
- Clarification on whether mechanistic relationships must be linear.
- Lack of computational efficiency details.
- Questions on graph scalability, deployment guidance, and interpretability.

### **Author Response Highlights**
- Clarified that nonlinear mechanistic models are supported.
- Discussed scalability limits for biomedical systems (typically ≤50 states).
- Reported detailed training-time comparisons.
- Added clinical examples (e.g., suppressed glucagon feedback).
- Revised manuscript to improve clarity.

### **Outcome**
Reviewer did not reply before the discussion freeze; **score remained 8**.

---

### **2. Reviewer mjPg — Score: 4 → 6 (increased)**

### **Main Concerns**
- Need stronger justification for RDAG transformation and Step 1 stability benefits.
- Step 2 (partial transitive closure) described too formally; missing intuition.
- Unclear how domain knowledge is incorporated.
- Minor typos and formatting issues.

### **Author Response Highlights**
- Added mathematical explanation of instability/stiffness caused by cycles, plus toy examples.
- Included intuitive “skip-grade” analogy for partial transitive closure.
- Clarified how domain knowledge can be injected at each reduction step.
- Corrected terminology (MSCC) and explained the spacing behavior.

### **Outcome**
Reviewer indicated improved clarity and **raised score from 4 → 6**.

---

### **3. Reviewer egCh — Score: 6 → 6 (unchanged)**

### **Main Concerns**
- Questioned rationale for DAG/RDAG structure without causal interpretation.
- L1/L2 regularization step seemed standard.
- Inconsistent y-axis scales made comparisons difficult.
- Concern about comparing MNODE (with priors) to black-box NODEs.

### **Author Response Highlights**
- Clarified role of RDAG in improving interpretability and stabilizing ODE dynamics.
- Justified novelty of the hybrid regularization linked to group LASSO structure.
- Added table versions of results for easier comparison.
- Discussed bias–variance trade-offs when using imperfect mechanistic priors.

### **Outcome**
Reviewer found responses satisfactory and kept positive score; **score remained 6**.

---

### **4. Reviewer 5nbh — Score: 4 → 6 (increased)**

### **Main Concerns**
- Handling of **patient heterogeneity**; fear model learns an “average patient.”
- Need confirmation of **patient-level** data splits to avoid leakage.
- Only retaining graph (not functional forms) seen as a weak mechanistic prior.
- TCN outperforming HGS at large N required discussion.

### **Author Response Highlights**
- Clarified that instance-specific initial conditions capture inter- and intra-patient variability.
- Provided empirical evidence that intra-patient variability ≈ inter-patient variability, mitigating leakage concerns.
- Added detailed discussion of hybrid modeling trade-offs (retain functional forms vs. residual learning vs. graph-only).
- Explained bias–variance behavior causing TCN to surpass HGS with large datasets.

### **Outcome**
Reviewer accepted resolutions and **raised score from 4 → 6**.

---

We sincerely thank the AC for the time and effort dedicated to overseeing the review process and for helping ensure a fair and constructive evaluation of our work.

Best Regards,

Authors

---

### Meta-Review · Area_Chair_bc6e · 2025-12-19

**Summary:**

This paper proposed a new pipeline to limit overfitting in hybrid neural ODEs with automatic state selection and structure optimization. While the paper initially received mixed scores, the reviewers had engaged with the authors in their response.

The concerns of Sy1g concerned largely discussion and interpretation, which the authors have addressed. Reviewer mjPg had mostly clarification questions, which were answered extensively with new results and discussion. I have to say I did not understand the causal part (also asked by egCh). In causality, usually cyclic dependencies are fine, usually just require specifying the interventional distributions explicitly. With time, it is often assumed that at any given time either A -> B or vice versa, with the cycle only appearing when aggregating. From a skim, it appears that this step is related to “causal abstractions”, which I encourage the authors to look at. The other significant concern was from 5nbh on patient variability, alongside other questions on the experiments (cross validation and data leakage). This was clarified by the authors to the reviewer satisfaction and the explanation makes sense to me.

Overall, the paper tackles an interesting problem, and while the connection with causality is not yet sufficiently clear (at least to me), I would recommend acceptance.

**Reviewer Concerns:**

As discussed above, the reviewers had a number of concerns that were addressed by the authors. I found one explanation not entirely satisfying, but the rest was good.

**Reviewer Scores:**

We have confirmation from the reviewers increasing their scores from the discussion, which is consistent with the response from the authors.

---

### Decision · Program_Chairs · 2026-01-26

Accept (Poster)